# DNA O-MAP uncovers the molecular neighborhoods associated with specific genomic loci

Yuzhen Liu[1,2†], Christopher D McGann[1†], Conor P Herlihy[1†], Mary Krebs[1], Thomas A Perkins[1], Rose Fields[1], Conor K Camplisson[1], David Z Nwizugbo[1], Qiaoyi Lin[1,3], Nicolas J Longhi[1], Chris Hsu[1], Shayan C Avanessian[1,2], Ashley F Tsue[3], Evan E Kania[3], David M Shechner[3,4,5], Brian J Beliveau[1,4,5]*, Devin K Schweppe[1,4,5]*

[1]Department of Genome Sciences, University of Washington, Seattle, United States; [2]Molecular and Cellular Biology Program, University of Washington, Seattle, United States; [3]Department of Pharmacology, University of Washington, Seattle, United States; [4]Brotman Baty Institute for Precision Medicine, Seattle, United States; [5]Institute of Stem Cell and Regenerative Medicine, University of Washington, Seattle, United States

*For correspondence:
beliveau@uw.edu (BJB);
dkschwep@uw.edu (DKS)

[†]These authors contributed equally to this work

## eLife Assessment

This study presents an **important** new method for probing the DNA and proteins associated with targeted chromatin domains in cells. The authors present **solid** evidence that the method can map DNA-DNA interactions for individual loci and can detect proteins enriched near repetitive DNA loci or targeted gene clusters. The methodological details of this study will be of particular interest and utility to chromatin biologists.

**Abstract** The accuracy of crucial nuclear processes such as transcription, replication, and repair depends on the local composition of chromatin and the regulatory proteins that reside there. Understanding these DNA–protein interactions at the level of specific genomic loci has remained challenging due to technical limitations. Here, we introduce a method termed 'DNA O-MAP', which uses programmable peroxidase-conjugated oligonucleotide probes to biotinylate nearby proteins. We show that DNA O-MAP can be coupled with label-free or sample multiplexed quantitative proteomics, targeted chemical perturbations, and next-generation sequencing to quantify DNA-proximal proteins and DNA–DNA interactions at specific genomic loci in human and murine cells. Furthermore, we establish that DNA O-MAP is applicable to both repetitive and unique genomic loci of varying sizes, from kilobase *HOX* gene clusters to megabase alpha-satellite repeats, and that DNA O-MAP can measure proximal molecular effectors in a homolog-specific manner.

## Introduction

Eukaryotic cells store their genetic material in the form of chromatin, a DNA–protein complex. The function of a eukaryotic DNA locus is executed through the cooperation between its nucleotide sequence and the hundreds of protein factors assembled around it. DNA–protein interactions thus play a fundamental role in regulating both the genome's structure and message storing functions (*Bickmore and van Steensel, 2013*). Therefore, developing methods to decipher DNA–protein interactions in cells has been a focus of technology development efforts for decades (*Jerkovic and Cavalli,*

*2021*). For instance, chromatin immunoprecipitation followed by sequencing (ChIP-seq; *Johnson et al., 2007*), which has emerged as a core technology for epigenomics (*Ho et al., 2012*), surveys the genome-wide binding profile of a target DNA-associated protein. ChIP-seq and related technologies e.g., DamID, *van Steensel and Henikoff, 2000*; CUT&Tag, *Kaya-Okur et al., 2019* have produced an abundance of high-quality datasets that enabled the establishment of database consortia such as ENCODE (*Luo et al., 2020*; *ENCODE Project Consortium, 2004*) and IHEC (*Bujold et al., 2016*), and significantly accelerated chromatin state annotation efforts (*Ernst and Kellis, 2012*; *Hoffman et al., 2012*). Such methods, which profile DNA–protein interactions through a protein-centric lens, require the *a priori* knowledge of which protein(s) to target and rely on the availability of suitable reagents such as antibodies or genetically engineered cell lines. By targeting a single protein at a time, these methods also inherently ignore the context of protein complexes or transient interactions that may be present at a given locus.

In addition to methods that profile the DNA bound by specific proteins, efforts have been dedicated to addressing the inverse problem—identifying the full collection of proteins assembled on a given DNA locus (*Gao et al., 2018*; *Myers et al., 2018*; *Qiu et al., 2019*; *Ugur et al., 2020*). Such methods include the foundational proteomics of isolated chromatin segment (PICh) technology, which uses a biotinylated oligonucleotide (oligo) probe to affinity label specific genomic DNA intervals via *in situ* hybridization (ISH) (*Dejardin and Kingston, 2009*). To enhance the stability of probe–chromatin interactions throughout the purification workflow, PICh utilizes oligos containing locked nucleic acid residues (*Silahtaroglu et al., 2003*), which are highly efficient as hybridization probes against repetitive DNA targets but cost-prohibitive to use to target non-repetitive intervals that require dozens to hundreds of probes to properly target a region of interest (*Beliveau et al., 2012*). As noted in follow-up work, PICh was effective for repeat sequences but would require significant additional work to extend to more complex genomic sequences (*Ide and Dejardin, 2015*). Additionally, even with the increased stability gained from the use of locked nucleic acid probes, the probe-chromatin hybrids can be difficult to maintain when coupled with stringent purification washes (*Ide and Dejardin, 2015*). As a consequence, an input of one trillion cells was used for a single purification and identification of proteins interacting with telomeres (*Dejardin and Kingston, 2009*).

To reach a higher degree of enrichment, which is critical for lower abundance DNA targets, an alternative strategy is to directly biotinylate the proteins that occupy a target DNA locus. This biotinylation can be achieved via targeted proximity labeling using promiscuous biotin ligases (*Roux et al., 2012*; *Cho et al., 2020*) or the engineered ascorbate peroxidase (APEX/APEX2) enzymes (*Lam et al., 2015*; *Martell et al., 2012*). Since the development of APEX, several methods including C-BERST (*Gao et al., 2018*) and GLoPro (*Myers et al., 2018*), have combined APEX with CRISPR genome targeting to endow it with locus specificity. This involves fusing APEX to a catalytically dead RNA-guided nuclease, Cas9 (dCas9), and directing the fusion enzyme to a specific locus of interest by single-guide RNAs (sgRNAs). The locus-docked dCas9-APEX biotinylates the neighboring proteins on electrophilic amino acid side chains, such as tyrosine, enabling protein purification and subsequent identification by mass spectrometry. In the case of GLoPro, APEX-based proximity labeling reduced the input required for each replicate analysis to ~300 million cells—a 10-fold reduction in cell input compared to PICh. Most recently, an approach termed TurboCas (*Cenik et al., 2024*) introduced the combination of dCas9 fused to the miniTurbo (*Branon et al., 2018*) biotin ligase, enabling detection of locus-specific proteins from 50 million cells per replicate. Nevertheless, a notable limitation of CRISPR-guided proximity labeling is the requirement of the fusion dCas9-APEX or dCas9-miniTurbo enzyme and sgRNAs in a suitable host cell line. Since a successful locus purification canonically requires tens to hundreds of millions of cells, if not more, most current methods aim to create stable cell lines for this purpose. These requirements limit the use of previous locus proteomics methods since efficient and well-tolerated gene delivery remains a major challenge and considerable effort in primary cells (*Mangeot et al., 2019*). In addition, the labeling reagents necessary for APEX-based proximity labeling—hydrogen peroxide and biotin phenoxyl radicals—are toxic to cells and living organisms, limiting the use of CRISPR-based peroxidase labeling to cell lines amenable to genetic engineering. Thus, an unmet need exists for extensible methods capable of scaling and profiling multiple genomic loci.

We address these technical limitations by introducing DNA O-MAP, a locus purification method that uses oligo-based ISH probes to recruit peroxidase activity to specific DNA intervals. DNA O-MAP

builds on our previously introduced RNA O-MAP (*Tsue et al., 2024*) and pSABER (*Attar et al., 2025*) techniques, which target peroxidase activity to specific RNAs and RNA/DNA intervals for purification or visualization, respectively. Here, we describe a cost-effective and scalable bulk hybridization and biotinylation workflows capable of processing millions of cells in parallel in just a few days and demonstrate that the recovered material is compatible with sample multiplexed proteomics (*Li et al., 2020*) and drug perturbation. We benchmark our approach by recovering telomere-specific DNA binding proteins after targeting telomeric DNA. We further showcase the scalability of DNA O-MAP by distinguishing the DNA-associated proteomes around human pericentromeric alpha-satellite repeats, telomeres, and mitochondrial genomes in quadruplicates using tandem mass tags (*Li et al., 2020*). We go on to demonstrate that DNA O-MAP can capture functionally relevant DNA–DNA interactions, read out by DNA sequencing, from 20 kb intervals. Additionally, we show that DNA O-MAP can measure the local proteome of non-repetitive elements such as the *HOXA* and *HOXB* gene clusters as well as differential proteomes at these gene clusters before and after chemical inhibition of chromatin regulation. Finally, we show that DNA O-MAP can be applied to discern homolog-specific local proteomes of the active and inactive X chromosome. We anticipate that the flexible targeting, scalable protocol, and robust labeling capabilities provided by DNA O-MAP will lead to its adoption as a platform technology for uncovering locus-proximal chromatin proteomes.

## Results

### Design of DNA O-MAP

DNA O-MAP is a molecular profiling methodology that combines the targeting flexibility of oligo-based ISH with the ability of horseradish peroxidase (HRP) to catalyze the localized deposition of small biomolecules at sites where they are bound. DNA O-MAP works by recruiting a 'secondary' HRP-conjugated oligo to sites where the primary ISH probes are bound. HRP-mediated deposition of biotin at targeted loci enables the pull-down and purification of proximal and chromatin-associated proteins and DNA from *trans*-interacting genomic loci. DNA O-MAP reports on both direct DNA–protein interactions and proteins in spatial proximity to a target locus and allows for identification and quantitative comparison of proximal proteins between target genomic loci. As in RNA O-MAP (*Tsue et al., 2024*), the specificity of ISH and/or biotinylation can be assessed by microscopy using a small sample of cells immobilized on solid support before the cells enter affinity purification. Importantly, the HRP-conjugated oligo is available via several commercial sources, allowing researchers without the expertise to perform their own conjugations to utilize DNA O-MAP.

### Establishing a scalable in-solution hybridization–biotinylation workflow for DNA O-MAP

During the development of DNA O-MAP, we refined an in-solution hybridization workflow on cells in suspension for cost-efficient genomic labeling in parallel with an in-dish workflow used for RNA O-MAP (*Tsue et al., 2024*; *Figure 1A*). We began with adherent cells grown on multi-layer flasks, each yielding 90–120 million cells, and subsequently released and fixed (4% PFA) in order to be compatible with DNA ISH. Samples can be processed in parallel, thereby increasing the number of samples that could be handled at once. We note that the in-solution version of the protocol reduces reagent costs by ~1000-fold relative to conventional ISH protocols performed on solid substrates to further enhance the scalability of DNA O-MAP.

### DNA O-MAP reveals the organization of the telomeric proteome

To demonstrate that O-MAP can successfully purify proteins from genomic viewpoints, we selected human telomeres for initial testing (*Figure 1A*). Mammalian telomeres are several kilobases of tandemly repeated arrays of 5′-TTAGGG-3′ hexamers with terminal 3′ single-stranded overhangs at the ends of chromosomes (*Chakravarti et al., 2021*). Telomeric DNA is specifically bound by a proteinaceous cap that protects the natural chromosome ends from being recognized as damaged DNA— the shelterin complex (*Sfeir and de Lange, 2012*; *de Lange, 2018*). Shelterin is a six-subunit complex, which is comprised of the telomeric repeat-binding factor 1 (TERF1), telomeric repeat-binding factor 2 (TERF2), protection of telomeres protein 1 (POT1), adrenocortical dysplasia protein homolog (ACD), TERF2-interacting protein 1 (TERF2IP), and TERF1-interacting nuclear factor 2 (TINF2). Due to the

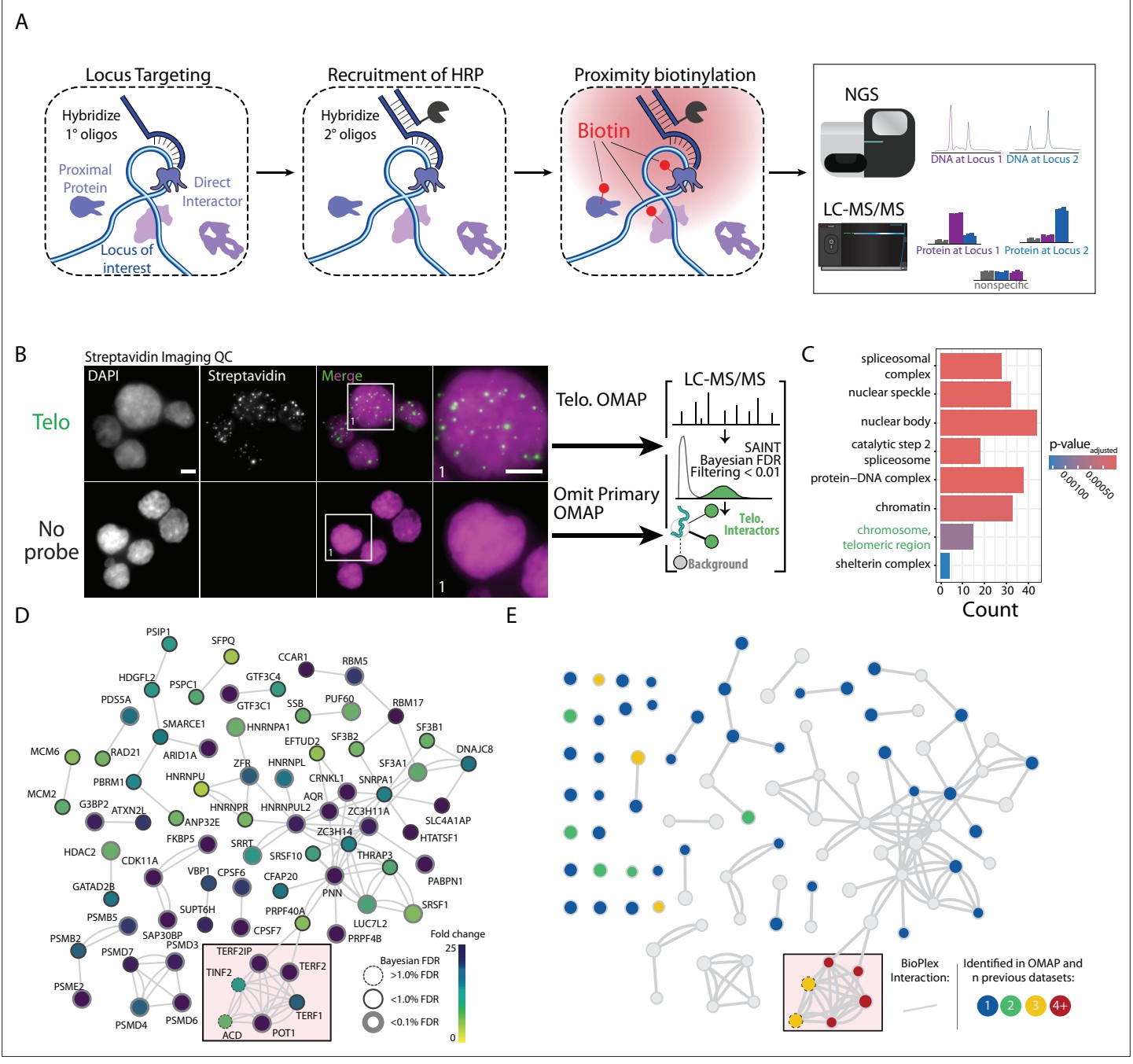

**Figure 1.** Overview of DNA O-MAP workflow and label-free quantitative proteomics analysis of telomeres. (**A**) Schematic of DNA O-MAP. (**B**) Fluorescent microscopy data showing the observed patterns of DNA (DAPI, left) and *in situ* biotinylation detected by staining with fluorescent streptavidin conjugates (middle, left) and overview of telomere-targeted DNA O-MAP experiment. (**C**) Significant gene sets identified by the gene set enrichment analysis of the proteins enriched by the telomere probe. (**D**) DNA O-MAP telomeric proteins mapped onto the BioPlex interaction network (*de Lange, 2018*; *Myung et al., 2004*). The red box highlights shelterin complex proteins. Nodes are colored by the fold-enrichment compared to a no-primary-probe control shown in B, excluding unconnected nodes. (**E**) Telomeric proteins observed in five previous datasets (PICh, C-BERST, CAPLOCUS, CAPTURE, and BioID) superimposed onto (E), colored by the number of prior datasets where the protein was present and including unconnected nodes. Scale bars, 5 μm.

unique telomeric sequence and characteristic DNA structure, the shelterin proteins accumulate at the ends of the chromosomes. Accordingly, this well-defined set of proteins has been widely accepted as goalposts for a successful locus-proximal enrichment experiment (*Gao et al., 2018*; *Myers et al., 2018*; *Dejardin and Kingston, 2009*). In the near-diploid HCT-116 cells, telomeres have an average

length of 5.6 kb and their cumulative length approximates 0.017% (~500 kb) of the human genome (*Supplementary file 4*; *Myung et al., 2004*). Compared to other repetitive elements in the human genome, telomeres are relatively short in HCT-116 cells and thus serve as a rigorous test case for DNA viewpoints of around 500 kb in aggregate across the genome.

We performed a DNA O-MAP experiment in which we either targeted telomeric DNA or omitted the primary hybridization probe (negative control). We purified biotinylated proteins from <60 million cells in three technical replicates followed by imaging of biotinylation and identification of proteins using label-free, MS1-based quantitative proteomics. By streptavidin staining, the punctate fluorescence pattern of biotin-labeled biomolecules closely mimicked telomere fluorescent *in situ* hybridization (FISH), whereas we did not observe patterns of these puncta in the negative control samples (*Figure 1B*). From our label-free proteomics analysis, we identified 163 proteins as significantly enriched at telomeres. As expected, gene set enrichment analysis (*Subramanian et al., 2005*) identified significant enrichment of telomeric chromosomal components, chromatin, and protein–DNA complexes (*Figure 1C, D*). Importantly, we identified all six shelterin proteins in the telomere sample, and these proteins were completely absent from the control samples. Of the six shelterin proteins, four (TERF1, TERF2, TERF2IP, and POT1) passed stringent false discovery rate control while ACD and TINF2 did not due to low spectral intensity.

To benchmark DNA O-MAP, we compared the full set of telomeric proteins to proteins observed in five established telomeric datasets (PICh, C-BERST, CAPLOCUS, CAPTURE, and BioID) (*Gao et al., 2018*; *Qiu et al., 2019*; *Dejardin and Kingston, 2009*; *Liu et al., 2017*; *Garcia-Exposito et al., 2016*; *Figure 1E*). DNA O-MAP captured both previously observed telomeric interacting proteins (shelterins) as well as telomere-associated proteins (ribonucleoproteins). We identified multiple heterogeneous nuclear ribonucleoproteins (hnRNPs) previously annotated as telomere-associated, including HNRNPA1 and HNRNPU. HNRNPA1 has been demonstrated to displace replication protein A (RPA) and directly interact with single-stranded telomeric DNA to regulate telomerase activity (*LaBranche et al., 1998*; *Zhang et al., 2006*; *Flynn et al., 2011*). HNRNPU belongs to the telomerase-associated proteome (*Fu and Collins, 2007*; *Izumi and Funa, 2019*) where it binds the telomeric G-quadruplex to prevent RPA from recognizing chromosome ends. We mapped DNA O-MAP enriched telomeric proteins to the BioPlex protein interactome (*Schweppe et al., 2018*; *Huttlin et al., 2021*) and observed that in addition to capturing proteins from previously observed telomeric datasets (*Figure 1E*), DNA O-MAP enriched for protein–protein interactors with previously observed telomeric proteins. Previous data found RBM17 and SNRPA1 at telomeres, and in BioPlex, these proteins interact with three SF3 proteins (SF3A1, SF3B1, and SF3B2). Though they were not identified in previous telomeric proteome datasets, all three of these SF3 proteins were enriched in the DNA O-MAP telomeric data. Furthermore, through interactions with G-quadruplex binding factors, these SF3 proteins are regulators of telomere maintenance (*Wang et al., 2016*).

## DNA O-MAP quantitatively compares nuclear- and mitochondrial-targeted DNA-proximal proteomes

We next evaluated the utility of DNA O-MAP to quantitatively measure proteins associated with specific genomic loci. We integrated sample multiplexing quantitative (*Li et al., 2020*; *Schweppe et al., 2020*; *Navarrete-Perea et al., 2018*; *Schweppe et al., 2019*) proteomics downstream of DNA O-MAP to enable spectral quantification of all samples simultaneously (*Figure 2A*). In our experimental design, we measured the proteomes at three well-characterized DNA loci with distinct protein occupants in the human genome and a no-primary probe negative control: (1) telomeres, (2) pericentromeric alpha-satellite repeats, (3) the mitochondrial genome, and (4) no primary probe negative control (*Figure 2B*). Centromeres are epigenetically defined chromosomal loci where kinetochore proteins assemble for spindle microtubule attachment to ensure equal chromosome segregation during cell division (*McKinley and Cheeseman, 2016*; *Talbert and Henikoff, 2022*). Human centromeres are located within the AT-rich alpha-satellite repeats, which are higher-order repeats composed of 171-bp monomeric units (*McNulty and Sullivan, 2018*; *Altemose et al., 2022*). Due to the sequence independence of centromeres, we utilized a previously described probe (*Attar et al., 2025*; *Deng and Beliveau, 2022*) that targets a subset of alpha-satellite repeats to represent centromeres, hereafter denoted as the 'Pan Alpha' probe (*Supplementary file 4*). The predicted genome-wide binding profile (*Aguilar et al., 2024*) of the pan-alpha probe closely overlaps with centromeres and covers

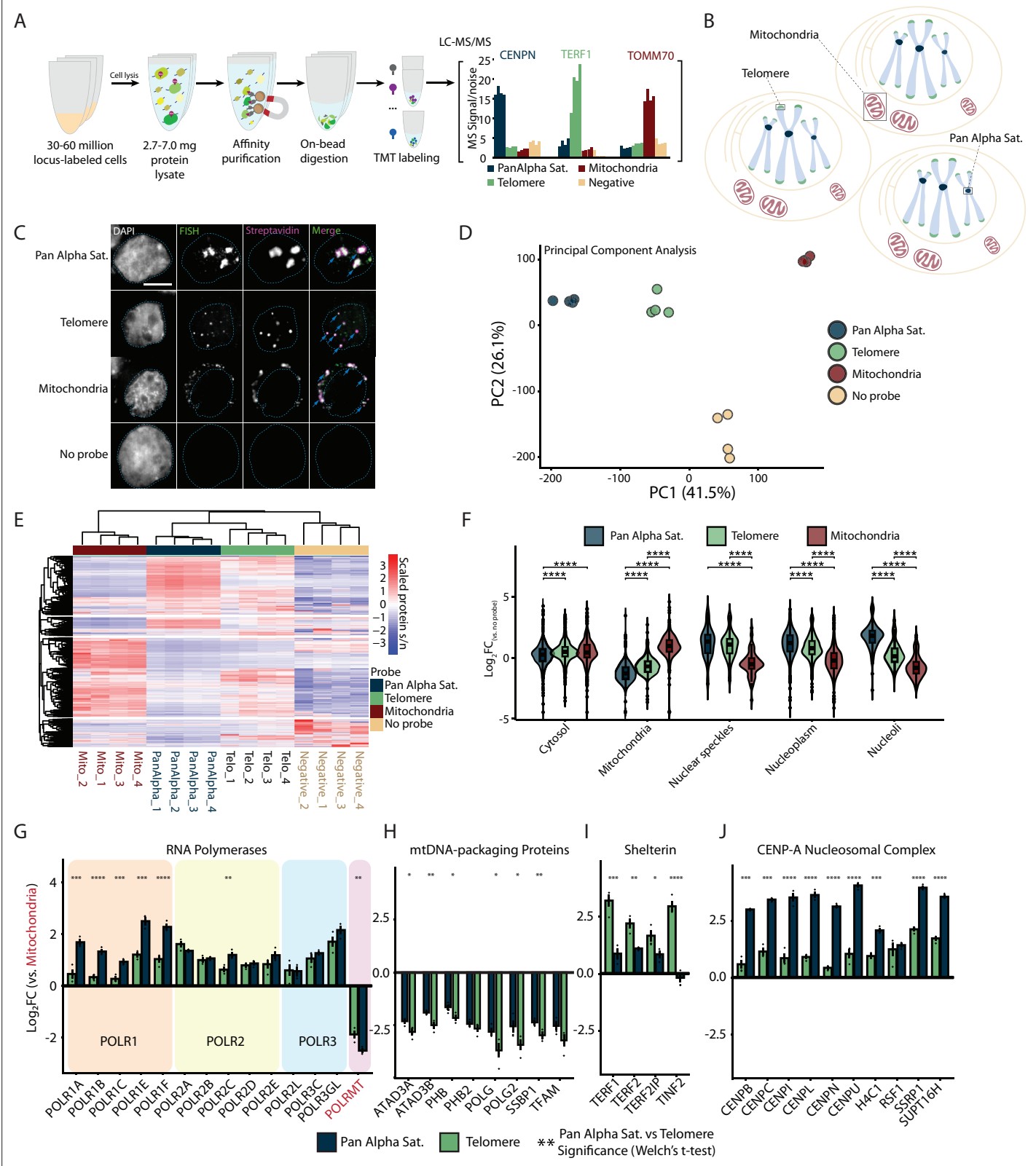

**Figure 2.** DNA O-MAP reveals distinct features of the sub-proteomes at pericentromeric alpha satellites, telomeres, and the mitochondrial genome. (**A**) Workflow of DNA O-MAP integrated with sample multiplexing quantitative proteomics. (**B**) Schematic of the three DNA loci examined in the TMT16plex experiment: pericentromeric alpha satellites, telomeres, and mitochondrial genomes. (**C**) Co-localization of DNA fluorescent *in situ* hybridization (FISH) and the streptavidin staining of the proteins biotinylated by DNA O-MAP targeting the pericentromeric alpha satellites, telomeres,

*Figure 2 continued on next page*

*Figure 2 continued*

and mitochondrial genomes. Scale bar: 5 µm. (**D**) Principal component analysis of scaled intensities of proteins enriched by the pan-alpha probe, telomere probe, mitochondrial genome oligo pool, and no-primary-probe control. (**E**) Unsupervised hierarchical clustering of scaled intensities of proteins enriched by the pan-alpha probe, telomere probe, mitochondrial genome oligo pool, and no-primary-probe control. (**F**) Log$_2$ fold change of proteins compared to no-primary-probe control, grouped by HPA subcellular location. Significance calculated based on Welch's *t*-test for pairwise comparisons (****p-value <0.0001). Log$_2$ fold change of proteins compared to mitochondrial probe enriched proteins for the RNA Polymerases (**G**), mtDNA nucleoid packaging proteins (*Matilainen et al., 2017*) (**H**), Shelterin (**I**), and CENP-A nucleosomal complexes (**J**). Significance calculated based on Welch's *t*-test for pairwise comparisons (p-value: *<0.05, **<0.01, ***<0.001, ****<0.0001).

The online version of this article includes the following figure supplement(s) for figure 2:

**Figure supplement 1.** Predicted genome-wide binding profile of the pan-alpha probe.

**Figure supplement 2.** Replicate analysis of multi-target DNA O-MAP proteomics experiment.

**Figure supplement 3.** Quantification of DNA O-MAP labeling specificity and efficiency for pan-alpha, telomere, and mitochondria probes.

**Figure supplement 4.** Relative quantitation for the multi-target DNA O-MAP proteomics experiment compared to no-probe control and mtDNA datasets.

**Figure supplement 5.** Comparison of histone proteins between telomere and pan-alpha probes.

an estimated 35 Mb (*Figure 2—figure supplement 1*). Mitochondria are intracellular organelles of eukaryotic cells with their own genome (mtDNA). The mtDNA is a circular double-stranded DNA molecule of about 16.6 kb, located in the mitochondrial matrix associated with the inner membrane (*Anderson et al., 1981*; *Rackham and Filipovska, 2022*). In HCT-116 cells, mtDNA copy number has been reported to range from 310 to 677 (*Zhou et al., 2020*).

To determine the localization of biotinylation using the new oligos and oligo pools, we performed DNA O-MAP in human HCT-116 cells with a co-hybridization of both fluorescent oligos and HRP oligos in order to observe FISH and *in situ* biotinylation signals in the same cell. Biotinylation patterns of the pan-alpha, telomere, and mtDNA probes showed strong concordance with FISH (*Figure 2C*). To quantify the local proteomes corresponding to each of these biotinylated patterns, we prepared replicate (*n* = 4) samples for each probe and control (*Figure 2—figure supplement 2*). Comparison of FISH and biotin signal revealed that DNA O-MAP labeling was highly specific to each of the telomeric, centromeric, and mitochondrial FISH signals (*Figure 2—figure supplement 3*). After *in situ* HRP-mediated labeling, we performed thermal reversal of fixation of cells prior to lysis, enrichment of biotinylated proteins (*Paek et al., 2017*), tryptic digestion, and labeling with isobaric TMTpro barcodes (*Li et al., 2020*). We confirmed that artifactual lysine acylation due to cellular fixation with PFA did not widely affect TMTpro labeling of peptides as only 1.38% of lysine-containing peptides were acylated with fixative. Notably, the fixative was not present during protease digestion, and thus all peptide N-termini were available for TMTpro labeling and quantitation, irrespective of lysine modification status.

In total, we quantified 3055 proteins across all four targeted and control samples (*Figure 2D, E*). We observed consistent proteome enrichment for replicate analyses with O-MAP by principal component analysis and correlation analyses (*Figure 2D, E*, *Figure 2—figure supplement 2*). Based on Human Protein Atlas annotations (*Thul et al., 2017*), we observed significant enrichment of mitochondrial proteins with the mtDNA-probe proteomes and proteins from nuclear locations such as nuclear speckles, nucleoplasm, and nucleoli enriched by the telomere and pan-alpha probes (*Figure 2F*, *Figure 2—figure supplement 4*). Notably, the pan-alpha probe enriched proteins from the nucleoli, consistent with the known nucleoli-centromere associations (*Bersaglieri et al., 2022*) and chromosomal passenger complex member AURKB, consistent with the centromeric localization of AURKB in early mitosis to ensure faithful chromosome segregation (*Liang et al., 2020*; *Broad et al., 2020*) and the localization of chromosomal passenger complex members to pericentromeric heterochromatin (*Rangasamy et al., 2003*; *Ono et al., 2004*). We also observed pericentromeric enrichment of spindle and chromosomal segregation associated proteins TPX2 (*Kufer et al., 2002*) and KIF20A (*Khongkow et al., 2016*; *Figure 2—figure supplement 4*, *Figure 2—figure supplement 5*).

Next, we explored the enrichment of several multi-unit protein complexes across the examined loci. To dissect the differences between enriched proteomes for each probe, we chose a subset of proteins of interest and measured the fold change of the two nuclear targets compared to mitochondria. RNA Polymerase I, II, and III subunits were all higher in the nuclear probes than mitochondria; however, in contrast to RNA Polymerases II and III, POLR1 proteins are significantly enriched in pan-alpha

compared to telomere (*Figure 2G*). This enrichment is likely due to clustering of centromeres around nucleoli (*Politz et al., 2013*; *Rodrigues et al., 2023*), the location of ribosomal RNA synthesis by RNA Polymerase I. Conversely, mitochondrial RNA Polymerase POLRMT abundance was significantly lower in the nuclear probe proteomes compared to the mitochondrial probe proteome ($\log_{2\ \text{Pan-Alpha Sat.}/\text{Mito.}}$ = −2.51; $\log_{2\ \text{Telomere/Mito.}}$ = −1.88). Similarly, we observed enrichment of mtDNA-packaging nucleoid components (*Matilainen et al., 2017*) with the mtDNA probes (TFAM, SSBP1, POLG, POLRMT, Lon, ATAD3A/B, and PHB/PHB2; *Figure 2G, H*). As above, we observed consistent enrichment of shelterin components at telomeres (*Figure 2I*). We also observed CENP-A nucleosomal complexes enriched in the pan-alpha proteomes (*Figure 2J*). Histones were enriched with our nuclear probes and a subset (H2A1C, H2AX, and H4C1) was significantly enriched by the pan-alpha probe compared to the telo-mere probe (*Figure 2—figure supplement 5*). We also observed enrichment of catenins CTNNB1 and CTNND1 at telomeres (*Figure 2—figure supplement 4*). The transcription factor CTNNB1 has been observed at the transcriptional start site of h*TERT* where it regulates h*TERT* expression (*Hoffmeyer et al., 2012*). The h*TERT* gene is located in the subtelomeric region of chromosome 5 (chr5:1,253,167–1,295,068) and expressed in HCT-116 cells (*Tsherniak et al., 2017*). Collectively, these results highlight the subcompartment sensitivity of DNA O-MAP to distinguish differential compartment components even for ubiquitous chromatin constituents like histones.

## DNA O-MAP uncovers DNA–DNA interactions from non-repetitive DNA loci

Beyond repetitive regions in the human genome, we explored whether DNA O-MAP can recover material from single-copy DNA intervals. To this end, we designed an experiment in which we performed *in situ* biotinylation followed by chromatin extraction, affinity purification, and sequencing (*Figure 3A*). The human genome is folded into thousands of chromatin loops where two loci on the same chromosome are tethered to each other (*Figure 3B*). The anchors of the loops are bound by the insulator protein CTCF. The ring-shaped cohesin protein complex is thought to often stall at CTCF-bound sites while dynamically moving along the genome, creating contact domains of preferential DNA–DNA interaction (*Rowley and Corces, 2018*). These contacts between chromatin loop anchors have been captured genome-wide with *in situ* Hi-C (*Rao et al., 2015*). Normally present in two copies per genome, these 20–25 kb loop anchor intervals are considerably less abundant than telomeres.

We first evaluated whether DNA O-MAP can specifically biotinylate loop anchors with micros-copy by a co-hybridization of both fluorescent oligos and HRP oligos at four anchors: chr3 left (chr3:187,729,712–187,749,712), chr3 right (chr3:188,939,711–188,964,711), chr19 left-2 (chr19:33,425,000–33,450,000), and chr19 right (chr19:33,750,000–33,775,000). DNA O-MAP specif-ically biotinylated the biomolecules proximal to these small DNA intervals, as observed in the co-lo-calizing patterns of FISH and streptavidin staining in the same cells (*Figure 3C*). We next evaluated whether DNA O-MAP could recover the DNA interactions originally discovered by Hi-C. We targeted a pair of intervals with high contact frequency—chr3 left and chr3 right anchors, one non-looping interval (chr10:123,187,984–123,207,984), and a no-primary-probe control. We performed DNA O-MAP to biotinylate these DNA intervals, subjected the labeled cells to chromatin solubilization and desthiobiotin purification, and sequenced the eluate DNA. As expected, all three probed DNA intervals were highly enriched compared with other genomic regions, indicating efficient purification of the loci (*Figure 3D*, *Figure 3—figure supplement 1*). Furthermore, chr3 left and chr3 right anchors reciprocally recovered each other, indicating that DNA O-MAP was able to recover known DNA inter-actions mediated by proteins. In contrast, the non-looping chr10 anchor did not enrich any other peak other than itself (*Figure 3—figure supplement 1B*). Lastly, in the cells that received no primary oligos, no pronounced enrichment was observed genome wide (*Figure 3—figure supplement 1B*).

To examine the multiplexability and reproducibility of DNA O-MAP, we simultaneously targeted three chromatin loop anchors: chr3 left, chr10 right (chr10:123,957,984–123,977,984), and chr19 right anchors in duplicates and subjected the cell pellets to purification and DNA sequencing. All three targeted anchors, chr3 left, chr10 right, and chr19 right anchors, were successfully enriched (*Figure 3E, F*, *Figure 3—figure supplement 2A*), whereas no pronounced enrichment was observed in the no-primary-probe controls genome-wide (*Figure 3—figure supplement 2B*). Furthermore, chr10 left (contacting chr10 right), chr19 left-1, and chr19 left-2 (both contacting chr19 right) were also effi-ciently recovered, accurately matching the Hi-C contact maps and the signals from two replicates was

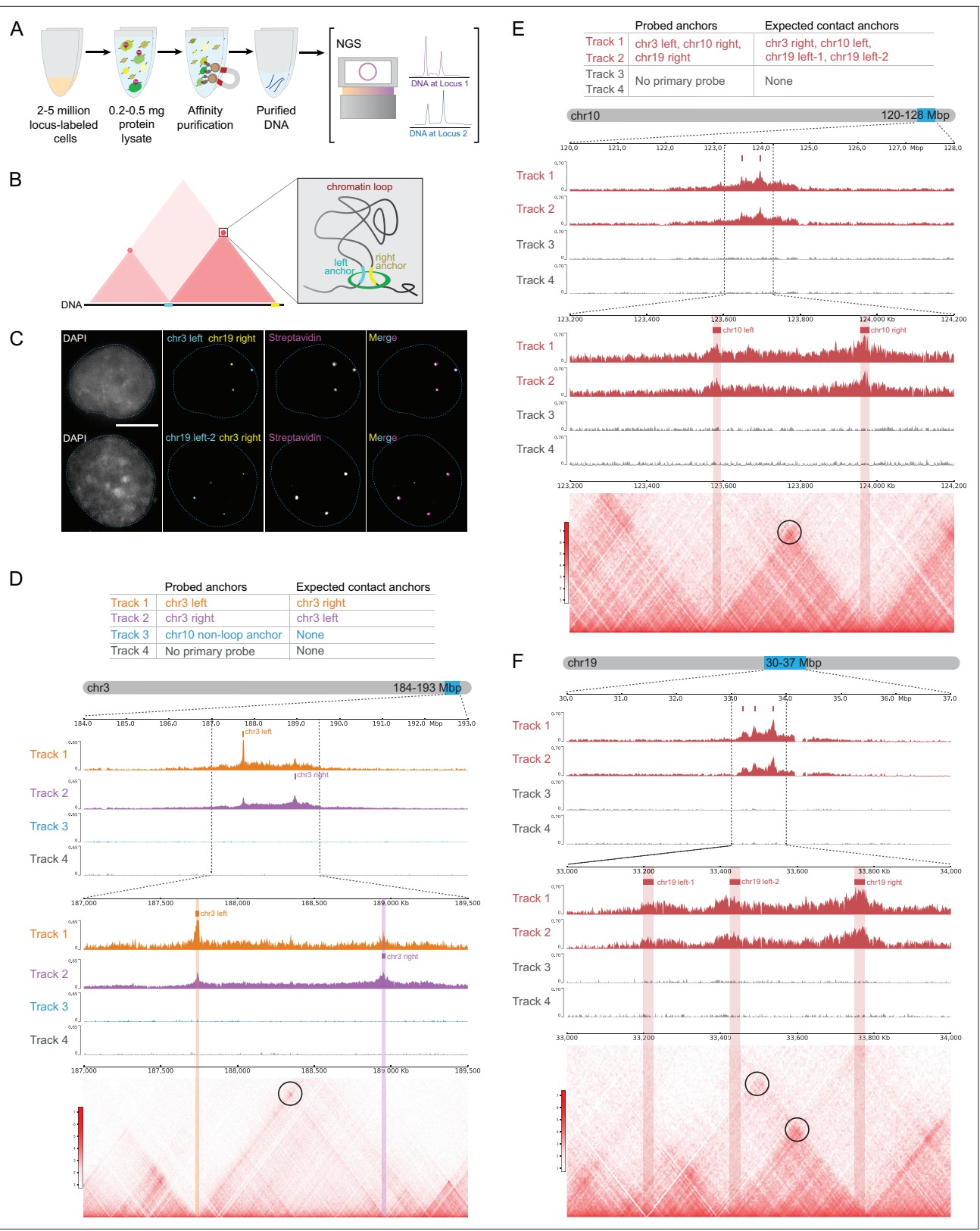

**Figure 3.** DNA O-MAP efficiently labels single-copy chromatin loop anchors. (**A**) Workflow of DNA O-MAP integrated with biotin purification sequencing. (**B**) Schematic of a pair of chromatin loop anchors on a hypothetical Hi-C map and three-dimensional space. (**C**) DNA fluorescent *in situ* hybridization (FISH) and the streptavidin staining of the proteins biotinylated by DNA O-MAP targeting anchors of chromatin loops on chromosome 3 and chromosome 19. (**D**) Table listing the three anchors (Tracks 1–3) and no-primary-probe control (Track 4) biotinylated by DNA O-MAP and their

*Figure 3 continued on next page*

*Figure 3 continued*

expected anchors in contact in each track (top). Desthiobiotin purification sequencing signals across the 9 Mb region on chromosome 3 corresponding to the chr3 chromatin loop (middle). Desthiobiotin purification sequencing signals and pairwise contact map at 5 kb resolution across the 2.5 Mb region on chromosome 3 corresponding to the chr3 chromatin loop. Black circle on the contact map indicates the presence of a loop (bottom). (**E**) Table listing the three chromatin loop anchors (Tracks 1 and 2) and no-primary-probe controls (Tracks 3 and 4) biotinylated by DNA O-MAP in duplicates and their expected anchors in contact in each track (top). Desthiobiotin purification sequencing signals across the 8 Mb region on chromosome 10 corresponding to the chr10 chromatin loop targeted (middle). Desthiobiotin purification sequencing signals and pairwise contact map at 5 kb resolution across the 1 Mb region on chromosome 10 corresponding to the chr10 chromatin loop. Black circle on the contact map indicates the presence of a loop (bottom). (**F**) Desthiobiotin purification sequencing signals across the 7 Mb region on chromosome 19 corresponding to the chr19 chromatin loops targeted (top). Desthiobiotin purification sequencing signals and pairwise contact map at 5 kb resolution across the 1 Mb region on chromosome 19 corresponding to the chr19 chromatin loops. Black circles on the contact map indicate the presence of loops (bottom).

The online version of this article includes the following figure supplement(s) for figure 3:

**Figure supplement 1.** DNA O-MAP biotin purification sequencing of chr3 left, chr3 right, chr10 non-loop anchors, and no-primary-probe control.

**Figure supplement 2.** DNA O-MAP biotin purification sequencing of multiplexed targeting of chr3 left, chr10 right, chr19 right anchors, and no-primary-probe control in duplicates.

**Figure supplement 3.** DNA O-MAP biotin purification sequencing enrichment score distribution of multiplexed targeting of chr3 left, chr10 right, and chr19 right anchors.

consistent (*Figure 3E, F*, *Figure 3—figure supplement 3*). We confirmed this by distance-dependent normalization previously developed for proximity-labeling methods (*Figure 3—figure supplement 3*; *Chen et al., 2018*). These imaging and genomics data demonstrate that DNA O-MAP is capable of labeling small, single-copy DNA intervals with high specificity.

## DNA O-MAP uncovers differential protein enrichment from non-repetitive loci

Given our success with recovering DNA–DNA contact information from unique loci, we explored whether DNA O-MAP can recover the local proteome from single-copy DNA intervals as we did earlier with repetitive intervals targeting telomeres, alpha-satellite repeats, and the mitochondrial genome. To investigate this, we utilized the *HOXA* and *HOXB* gene clusters as targets. The *HOX* subgroup of the homeobox family of genes has been well described for their roles in determining body plan formation, cell identity, and has been implicated in susceptibility to oncogenesis (*Steens and Klein, 2022*; *Pinto et al., 2024*; *Hubert and Wellik, 2024*; *Shah and Sukumar, 2010*; *Khan et al., 2024*). Here we designed probes to target 83 and 81 kb for *HOXA* and *HOXB*, respectively. We performed DNA O-MAP targeting either *HOXA*, *HOXB*, or no primary hybridization probes from 50 million human K562 cells across six replicates for *HOXA* and *HOXB* and five replicates for no primary (*Figure 4A*). We also performed DNA O-MAP in human K562 cells with co-hybridization of both fluorescent oligos and HRP oligos to demonstrate locus-targeted biotinylation and FISH signal within the same cell. Biotinylation patterns for both *HOXA* and *HOXB* showed strong concordance with the FISH signal (*Figure 4B*). Identification of biotinylated proteins was carried out using label-free quantitative proteomics. From our label-free proteomics analysis, we identified 42 proteins that were significantly differentially enriched between the *HOXA* and *HOXB* loci (*Figure 4C*, *Figure 4—figure supplement 1*). Of the differentially enriched proteins, both HDAC3 and TCF12 were scored as enriched in the *HOXB* labeled sample compared to *HOXA*. This agrees with ENCODE (*Zhang et al., 2020*) ChIP-seq data in K562 cells, with more called peaks for these proteins at *HOXB* than *HOXA* (*Figure 4D*). Additionally, SMARCB1 was scored as enriched in the *HOXA* labeled sample compared to *HOXB* and ENCODE (*Zhang et al., 2020*) ChIP-seq data showed enrichment of SMARCB1 at *HOXA* over *HOXB* (*Figure 4D*). SWI/SNF factors such as SMARCB1 have been described to interact with the *HOX* genes (*Weber et al., 2021*), but differential enrichment has not been well documented.

To further test the sensitivity and application of our non-repetitive DNA O-MAP approach at the *HOX* genes, we performed a chemical perturbation analysis targeting *HOXA* and *HOXB* in the presence or absence of enhancer of zeste homolog 2 (EZH2) inhibition with GSK126. In differentiated cells, such as K562s, the *HOX* genes are silenced by the Polycomb Repressive Complex 2 (PRC2). The main catalytic component of PRC2 is EZH2, which is responsible for trimethylating histone 3 at lysine 27 (H3K27me3) (*Cao et al., 2002*; *O'Meara and Simon, 2012*). Deposition of H3K27me3 has been well characterized as a mark of transcriptional repression associated with PRC2 activity. Additionally,

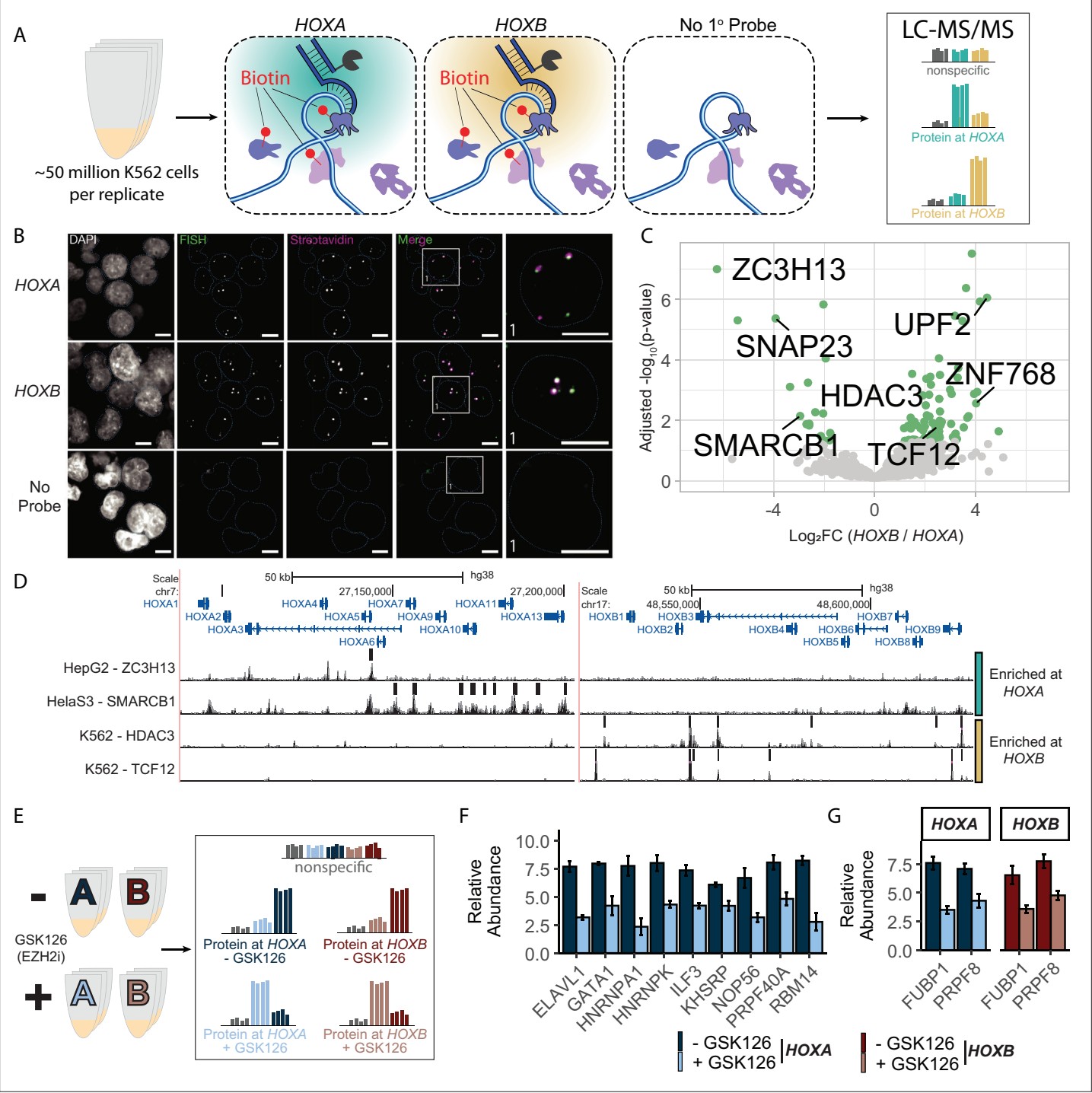

**Figure 4.** DNA O-MAP efficiently identifies the local proteome of the *HOXA* and *HOXB* gene clusters. (**A**) Schematic of DNA O-MAP being applied to the *HOXA* and *HOXB* gene clusters for identification of differentially enriched proteins. (**B**) Representative images depicting overlap of fluorescent *in situ* hybridization (FISH) and streptavidin labeling at *HOXA* and *HOXB* loci. (**C**) Volcano plot of proteins identified at *HOXA* and *HOXB* loci. Each dot represents a single protein with proteins of interest called out in black. Green dots indicate proteins that passed significant enrichment thresholds with an absolute Log$_2$ Fold Change greater than 1 (twofold change) and corrected p-value <0.05. (**D**) ENCODE ChIP-seq data showing peak calls and p-values at *HOXA* and *HOXB* loci for selected enriched proteins, ZC3H13, SMARCB1, HDAC3, and TCF12. (**E**) Schematic depicting the use of GSK126 with DNA O-MAP. (**F**) Bar chart showing proteins with significantly altered abundance following treatment with GSK126 at *HOXA*. (**G**) Bar chart showing proteins with significantly altered abundance at both *HOXA* and *HOXB* following treatment with GSK126 (Welch's *t*-test, corrected p-value <0.05).

The online version of this article includes the following figure supplement(s) for figure 4:

*Figure 4 continued on next page*

*Figure 4 continued*

**Figure supplement 1.** DNA O-MAP elucidates differences between the *HOXA* and HOX*B* proximal proteomes and changes to their proteomes following inhibition of EZH2 with GSK126.

EZH2 inhibition has become a prominent target in cancer therapies (*Zeng et al., 2022*; *Guo et al., 2024*). For the perturbation analyses, we used sample multiplexing quantitative proteomics (*Li et al., 2020*; *Schweppe et al., 2020*; *Navarrete-Perea et al., 2018*) and DNA O-MAP to quantify the local proteomes in response to drug inhibition (*Figure 4E*). We prepared replicate ($n$ = 3) samples for each probe following treatment with either the EZH2 inhibitor GSK126 or DMSO control. In total, we identified 11 proteins at *HOXA* and 8 proteins from *HOXB* with significantly altered enrichment following EZH2 inhibition (*Figure 4F, G*, *Figure 4—figure supplement 1B*), many of which are RNA-interacting proteins, such as ELAV1, RMB14, ILF3, HNRNPK, HNRNPA1, PRPF40A, and PRPF8. These data are consistent with recent evidence that suggests that long non-coding RNA-interacting proteins are critical for the establishment of PRC2-repressed loci (*Jansz et al., 2018*; *Pintacuda et al., 2017*). Several proteins we identified also have established connections to PRC2 repression of the *HOX* genes, such as GATA1, which has been suggested to mediate the switch to non-canonical PRC2 function (*Xu et al., 2015*). These data demonstrate that DNA O-MAP has sufficient sensitivity to detect proteomes at non-repetitive genomic loci and measure inhibitor-induced changes to the proximal proteome at sub-megabase, non-repetitive genomic regions (~80 kb).

## Establishing an on-plate hybridization–biotinylation workflow for DNA O-MAP

During the development of DNA O-MAP, we refined an on-plate workflow on cells adhered to glass bottom plates (*Figure 5A*). We began with adherent cells grown in 6-well plates, using 3 wells per replicate per condition with each well yielding 0.75–1 million cells. The cells are subsequently fixed (4% PFA) in order to be compatible with DNA ISH. Multiple plates can be processed in parallel to enable multiple replicates per experiment. We note that the on-plate version of the protocol further reduces the total number of cells required per replicate by ~10-fold compared to the in-solution protocol.

## DNA O-MAP detection of homolog-specific proteomes

To further understand the level of sensitivity we can achieve when investigating the local proteome of non-repetitive loci, we explored whether DNA O-MAP can recover material in a homolog-specific manner. To this end, we designed primary hybridization probes that differentially target either the active or inactive X chromosome homolog (Xa and Xi, respectively). The Xa–Xi paradigm is a particularly interesting variable homolog target as the Xi is largely transcriptionally silenced through cooperative activity of the non-coding RNA Xist and H3K27me3 by the PRC, along with other factors, in direct opposition of the Xa (*Jansz et al., 2018*; *Pintacuda et al., 2017*; *Bousard et al., 2019*; *Markaki et al., 2021*; *Wang et al., 2019*). Thus, these two homologs present a vastly different chromatin environment. To individually target Xi and Xa, we used differential single-nucleotide variants (SNVs) between the maternal and paternal homologs in EY.T4 female mouse fibroblast cells that were identified by homolog resolved genome sequencing (*Yildirim et al., 2012*; *Beliveau et al., 2015*). In doing so, we were able to design two sets of primary hybridization probes to target the same 4.5 Mb region on either Xi or Xa straddling the X inactivation center (*Figure 5B*). The approach of leveraging SNVs for homolog-specific probe binding has been previously validated for use with other ISH-based methods (*Beliveau et al., 2015*; *Nir et al., 2018*), but has not been previously applied to proximity labeling or proteomics approaches.

For validation of our differential labeling approach, we performed FISH for Xist and Xa alongside streptavidin labeling of Xi. Labeling patterns of Xi and Xist overlapped as expected while remaining spatially distinct from Xa FISH (*Figure 5C*). We performed a DNA O-MAP experiment in which we targeted either Xa or Xi across four replicates. Each replicate consisted of three wells of a 6-well plate each containing 0.75–1 million cells (*Figure 5A*). Of note, in contrast to the previous experiments, here we did not release our adherent cells and all steps were carried out with cells fixed to glass-bottom 6-well plates prior to lysis. From our label-free proteomics analysis, we identified 96 proteins that were significantly differentially enriched at either Xi or Xa (*Figure 5D*, *Figure 5—figure supplement*

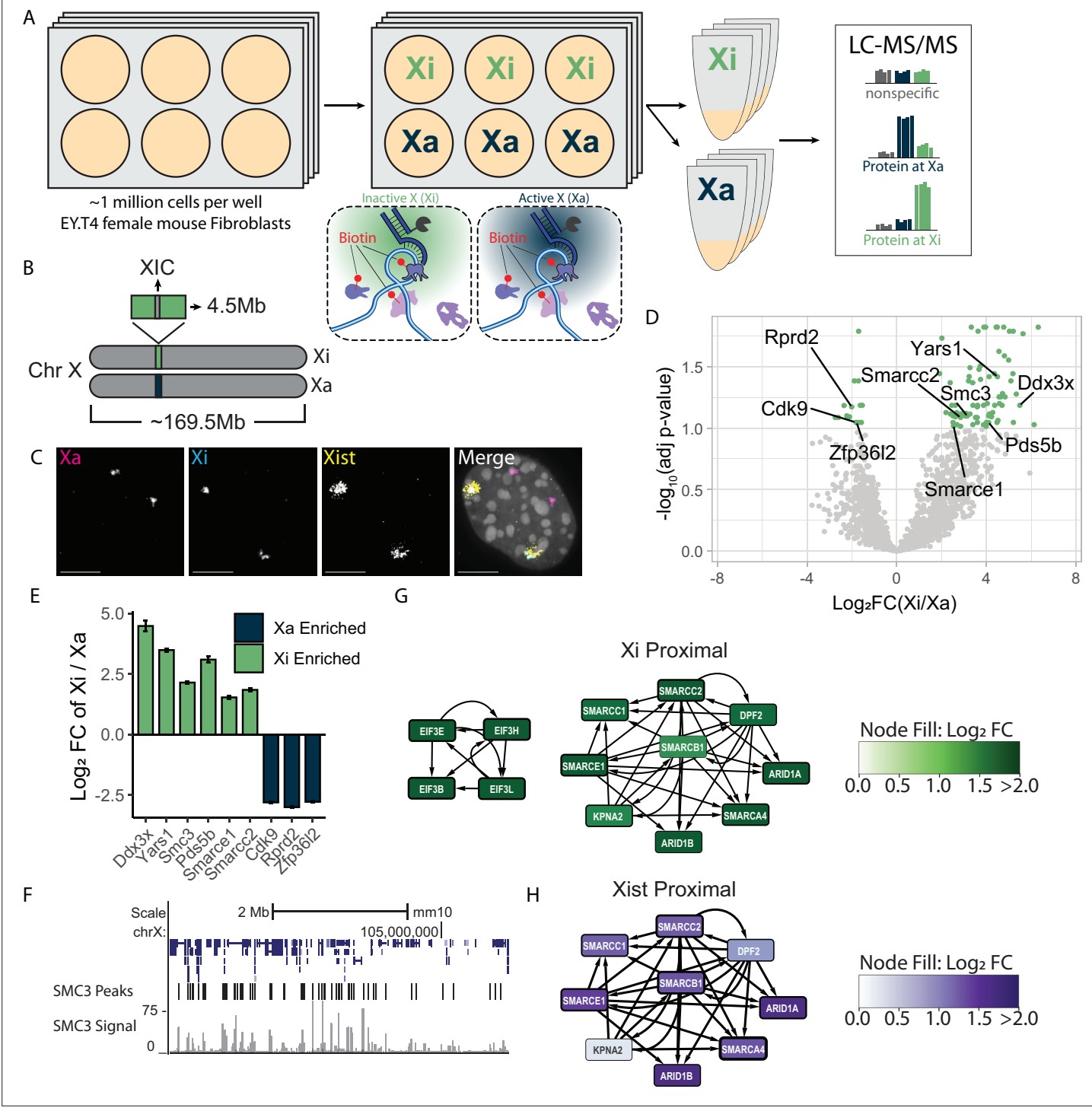

**Figure 5.** DNA O-MAP elucidates the homolog-resolved chromosome X proteome. (**A**) Schematic of DNA O-MAP being applied to Xi and Xa for identification of differentially enriched proteins. (**B**) Schematic showing the region of the X chromosome targeted by our primary hybridization probes. (**C**) Representative images depicting overlap of Xist fluorescent *in situ* hybridization (FISH) and Xi streptavidin labeling while spatially differentiated from Xa FISH. Scale bars are 10 µM. (**D**) Volcano plot of proteins identified at Xa and Xi. Each dot represents a single protein with proteins of interest called out in black text. Green dots indicate proteins that passed significant enrichment thresholds with an absolute Log$_2$ Fold Change greater than 1 (twofold change) and corrected p-value <0.1. (**E**) Bar chart showing example proteins with significant enrichment at Xi in green and at Xa in blue. Corrected p-value <0.1. (**F**) ENCODE ChIP-seq data in mouse fibroblast cells at our targeted region of chromosome X for SMC3. (**G**) Protein interaction networks of EIF and SWI/SNF complexes enriched at Xi. Node width is a function of corrected p-value and node color is a function of enrichment (Log$_2$ Fold

*Figure 5 continued on next page*

Figure 5 continued

Change). (**H**) Protein interaction network of the SWI/SNF complex from previously published RNA O-MAP of Xist. Node width is a function of −Log$_{10}$(p-value), and node color is a function of enrichment (Log$_2$ Fold Change).

The online version of this article includes the following figure supplement(s) for figure 5:

**Figure supplement 1.** DNA O-MAP elucidates the differential proteomes of the X chromosome at homolog resolution.

*1*). The Xa showed enrichment of proteins associated with a more active transcriptional pattern in comparison to Xi, such as RPRD2, ZFP36l2, and CDK9 (*Figure 5D, E*). For the Xi in comparison to Xa, we observed enrichment of protein interactors and complexes (*Schweppe et al., 2018*; *Huttlin et al., 2021*), including the EIF3 family, members of the cohesin complex Smc3 and Pds5b, and SWI/SNF factors. Both the cohesin and SWI/SNF complexes have been previously shown to interact with the inactive X chromosome (*Minajigi et al., 2015*) and enrichment of SWI/SNF factors was consistent with recent work measuring the proteome of Xist, the Xi-resident long non-coding RNA (*Tsue et al., 2024*; *Figure 5G,H*). Interestingly, we also see enrichment of Ddx3x, an X-linked gene previously implicated as a mediator of sex-based differences in neurodevelopment and disease (*Mossa et al., 2025*). Here, we show that DNA O-MAP was able to obtain homolog-specific, differentially enriched local proteomes from a 4.5 Mb, non-repetitive, region on the X chromosome from less than 3 million cells per replicate.

## Discussion

By combining the versatility of hybridization-based genome targeting with the robustness of proximity biotinylation, DNA O-MAP offers a scalable approach to study DNA-proximal proteomes of a specific locus. The hybridization–biotinylation workflow allows for efficient processing of samples and is compatible with both proteomic and genomic readouts. Integration with multiplexed quantitative proteomics enables simultaneous analysis of multiple loci or conditions, increasing data completeness and throughput. Label-free analysis of the telomeres shows strong concordance of labeling with ISH and recapitulates previous similar proteomic datasets. Our tri-locus experiment was able to differentiate proteins with a quantitative profile suggesting general nuclear location from those specifically associated with telomeres and pericentromeres. DNA O-MAP can target single-copy loci, as evidenced by the chromatin loop anchor, *HOXA* and *HOXB* gene cluster, and X chromosome targeting experiments, which enables the study of a wide range of DNA-proximal proteomes. The ability to detect differences in these proteomes is exemplified by the application of DNA O-MAP to the *HOXA* and *HOXB* loci following EZH2 inhibition and to the X chromosome in a homolog-specific manner.

O-MAP has now been shown to be a highly flexible technology for the exploration of biomolecular interactions with RNAs (*Tsue et al., 2024*) and DNA loci. Using oligos to target the DNA locus, DNA O-MAP can be theoretically adapted for use in any sample type amenable to ISH, including cultured cells, tissue sections, and primary tissue samples (*Attar et al., 2025*; *Aguilar et al., 2024*; *Hershberg et al., 2021*). As the purification tag is decoupled from the probe oligos, labeled chromatin fragments can undergo stringent washes to achieve efficient purification with minimal background. Moreover, without the need to genetically modify the biological system at hand, the probes in this dataset alone could be used to explore telomeric remodeling in cancer cells (*Garcia-Exposito et al., 2016*), spindle-associated proteome dynamics at the pericentromere (*Santos-Barriopedro et al., 2021*), and molecular drivers of hetero- or euchromatin formation (*Iglesias et al., 2020*) at nearly any locus in the human genome (O-MAP probes can feasibly cover >99% of the human genome) (*Aguilar et al., 2024*; *Hershberg et al., 2021*). Additionally, we show that this method is able to detect DNA–DNA contacts through detection of biotinylated loop anchors. Our approach functions similarly to 4C (*Simonis et al., 2006*); however, the biotin labeling of contacts does not rely on pairwise ligation events. Thus, detection of contacts through DNA O-MAP will vary in the sampling of DNA-DNA contacts in comparison.

DNA O-MAP has several current limitations. First, fixation can modify proteins and must be accounted for when labeling is performed in fixed cells. Second, DNA O-MAP captures proximal proteins for a given genomic locus. This means measured proteins will be both direct interactors and also proteins that are reproducibly within close physical proximity to the locus. This distinction has been described previously (*Tsue et al., 2024*; *Mathew et al., 2022*), but is generally important for interpretation of DNA O-MAP proteomics results and design of future DNA O-MAP experiments. This

is due, at least in part, to the long-lived nature of the biotin-phenoxy radical species generated by peroxidase-based proximity labeling methods. Third, DNA O-MAP may miss proteins at specific loci if the protein is especially low abundance, highly post-translationally modified, or otherwise challenging to detect by mass-spectrometry-based proteomics. Finally, DNA O-MAP reports more proteins than those that are 'specific' to a given locus. For example, all nuclear targeting probes identified nuclear, but non-specific, proteins like histones. An important aspect of the quantitative approaches used here is that proteins are measured as differentially abundant at a given loci in comparison to other loci or across treatments.

By taking a comparative quantitative approach, we remove the need to pre-define the local context of probe localization, but experimental design is critical and novel interactors may require further validation to confirm their co-localization at a given locus (e.g., with imaging/FISH). With developments in automation and instrument sensitivity, DNA O-MAP has the potential to expand to post-translational modifications and be used for large-scale chromatin perturbation screens. We anticipate that DNA O-MAP will have broad utility for research questions seeking to understand the intricate relationships between DNA sequence, chromatin structure, and cellular function.

## Methods
### Cell culture and fixation
Authenticated colorectal cancer HCT-116 cells were obtained from ATCC and grown in ATCC-formulated McCoy's 5A Medium Modified (ATCC 30-2007). EY.T4 mouse fibroblasts (a generous gift of the Jeannie T. Lee lab) were authenticated by *in situ* karyotyping and grown in ATCC-formulated DMEM (Gibco 11965-092). Authenticated human lymphoblast K562 cells were obtained from ATCC and grown in ATCC-formulated RPMI (Gibco 11875-093). All cell lines were supplemented with 10% fetal bovine serum and 100 U/ml penicillin–streptomycin and grown at 37°C in a humidified atmosphere of 5% $CO_2$. All cell lines tested negative for mycoplasma contamination in PCR-based assays. For each HCT 116 purification, 20 million cells were seeded into one T-500 flask (Thermo Scientific 132867) to culture for 36–48 hr to reach 90–120 million cells. Before collection, cells were briefly rinsed once with Dulbecco's phosphate-buffered saline (DPBS) and then incubated with 25 ml of TrypLE Express Enzyme (Gibco 12604-021) at 37°C for 2 min or until loosely attached. For each K562 purification, 25 million cells were seeded into one T-500 flask to culture for 48 hr to reach 100 million cells. Before collection, cells were briefly rinsed once with DPBS. For inhibition of EZH2, K562 cells were grown and harvested, as described above, in medium supplemented with either 5 µM GSK126 (Tocris 6790) or DMSO for 48 hr and then collected. For both HCT 116 and K562 purifications, the cell suspension was collected into two 50 ml conical tubes and the T-500 flask was rinsed with DPBS. The wash was combined with the cell suspension and centrifuged at 300 × *g* for 5 min. After a DPBS wash to remove remaining TrypLE, cells were fixed in 4% paraformaldehyde (wt/vol) (Electron Microscopy Sciences 15710) in phosphate-buffered saline (PBS) in suspension at room temperature for 10 min with rotation, followed by 125 mM Glycine quenching for 5 min at room temperature with rotation and 15 min on ice. Fixed cells were collected by centrifugation at 350 × *g* for 5 min and stored in fresh DPBS at 4°C until liquid-phase hybridization. Fixed cells were used within 3–5 days.

For each EY.T4 purification, 300,000 cells were seeded into each well of a glass-bottom 6-well plate (Cellvis P06-1.5H-N) to culture for 24 hr to reach 750,000–1 million cells per well. Cells were briefly rinsed once with DPBS and then fixed in 4% paraformaldehyde (wt/vol) (Electron Microscopy Sciences 15710) in PBS at room temperature for 10 min. Fixed cells were then washed with DPBS three times for 5 min each. Fixed cells were stored in fresh DPBS at 4°C until solid-phase hybridization. Fixed cells were used within 5 days.

### Primary oligo probes
Primary oligos targeting the human alpha-satellite repeat and telomere were purchased as individually column-synthesized DNA oligos from Integrated DNA Technologies. Probe sets targeting *HOXA* (chr7:27,092,311–27,175,959), *HOXB* (chr17:48,527,413–48,608,584), mtDNA (chrM:1–16,569), chr3 left anchor (chr3:187,729,712–187,749,712), chr3 right anchor (chr3:188,939,711–188,964,711), chr10 non-looping anchor (chr10:123,187,984–123,207,984), chr10 right anchor (chr10:123,957,984–123,977,984), and chr19 right anchor (chr19:33,750,000–33,775,000) were designed using PaintSHOP

(*Hershberg et al., 2021*) and ordered in oPool format from Integrated DNA Technologies. Homolog-specific chromosome X (chrX:100,254,241–102,428,950, 102,601,850–104,777,629) targeting probes were designed using an in-house computational pipeline and ordered in oPool format from Integrated DNA Technologies. More than 300 primary oligos were designed to cover each single-copy DNA interval to ensure a sufficient number of probes at the locus for FISH. The sequences of the oligo and oligo pools used are listed in *Supplementary file 1*.

## Oligo library synthesis

Amplification of the *HOXA*, *HOXB*, Xi, and Xa targeting primary oligo libraries was performed as follows. 20 ng μl$^{-1}$ of oligo library in 10 mM Tris, pH 8.0 was amplified by PCR. The reaction mix contained 34 μl of dH$_2$O, 10 μl 5× Phusion HF Buffer, 1.5 μl of 10 mM dNTP Mix, 1.5 μl of 10 μM F primer, 1.5 μl of 10 μM R primer, 1.0 μl of resuspended oligo pool, and 0.5 μl of Phusion DNA Polymerase (2 U μl$^{-1}$). The thermal cycler program was as follows: 95°C for 3 min, followed by 12 cycles of 98°C for 20 s, 60°C for 15 s, 72°C for 15 s then 72°C for 1 min followed by a 4°C hold. This PCR product was purified using a Zymo DNA Clean and Concentrator-5 (DCC-5) kit according to the manufacturer's protocol. 20 pg μl$^{-1}$ of the first PCR product was prepared as a template for the second PCR. The second PCR mix contained 27 μl of dH$_2$O, 10 μl of 5× Phusion HF Buffer, 1.5 μl of 10 mM dNTP Mix, 5.0 μl of 10 μM F Primer, 5.0 μl of 10 μM R Primer, 1.0 μl of diluted DNA template and 0.5 μl of Phusion DNA Polymerase (2 U μl$^{-1}$). The thermal cycler program was repeated, but for 18 cycles instead of 12. The second PCR product was purified as before. RNA was synthesized using the NEB HiScribe T7 Quick High Yield RNA Synthesis Kit with a modified reaction mix containing 8 μl of dH$_2$O, 2.5 μl of diluted DNA template, 15.0 μl of NTP Buffer Mix, 3 μl of T7 RNA Polymerase Mix, and 1.5 μl of RNaseOUT. The reaction was carried out for 16 hr at 37°C followed by a 12°C hold. The DNA template was then digested using DNase I (M0303) by adding 2 μl (4 units) of DNase I and 50 μl of Ultra Pure Water (UPW) (10977-015) to each reaction. The reverse transcription reaction contained 45 μl of synthesized RNA, 15 μl of 5× RT Buffer, 10.5 μl of 10 mM dNTP Mix, 1 μl of 100 μM RT Primer, 1.5 μl of RNaseOUT and 2 μl of Maxima H Minus Reverse Transcriptase (200 U μl$^{-1}$) and 20 μl of UPW for a total volume of 75 μl. The reactions were incubated at 50°C for 2 hr, 85°C for 5 min, then a 4°C hold. RNA templates were degraded by adding 37.5 μl of 0.5 M EDTA and 37.5 μl of 1 M NaOH to each reaction and incubated at 95°C of 10 min followed by a 4°C hold. Final ssDNA probe was purified using a Zymo DNA Clean and Concentrator-100 (DCC-100) kit according to the manufacturer's protocol for oligo purification.

## Primer exchange reaction

To extend primary oligos with primer exchange reaction (PER) concatemers, reactions were set up as previously described (*Kishi et al., 2018*) in 100 μl-volume containing 10 mM MgSO$_4$, 300 μM dATP/dCTP/dTTP mix, 100 nM Clean.G hairpin, 80 U/ml Bst DNA Polymerase, Large Fragment (NEB M0275L), 1 μM hairpin, and 1 μM primary oligos in PBS. To verify the length of primary oligos, the reactions were assessed with denaturing polyacrylamide gel electrophoresis. Primary oligos extended to 300–500 nucleotides were used in hybridizations downstream. Unpurified reactions were dehydrated using vacuum concentrators and stored dry at –20°C until hybridization.

## In-solution hybridization and biotinylation of cell pellets

HCT 116 and K562 oligo hybridizations were performed on cells in solution for the cost-effectiveness of primary and secondary oligos. Fixed cells were split into 6e7 cell aliquots in 1.5 ml microcentrifuge tubes. All washes and buffer exchanges were performed as follows: centrifuging at 350 × *g* for 3.5 min or until pelleted, pouring away used buffers from the pellets, adding new buffers, and gentle shaking or low-speed vortexing to dislodge cell pellets into tiny clusters or cell suspensions for incubations or washes. Cells in fresh wash buffer were rotated on a low-speed nutator for 5 min.

Cells were rinsed once with fresh PBS and permeabilized in PBS-0.5% Triton X-100 (Sigma T8787) for 10 min with nutation. After a PBS-0.1% Tween20 (PBS-T) (Sigma T2287) wash, permeabilized cells were incubated in 0.1 N hydrochloric acid (HCl) for 5 min. After a PBS-T wash to remove acid, cells were incubated in PBS-T-0.5% hydrogen peroxide to block endogenous peroxidases. After a 2X saline sodium citrate-0.1% Tween20 (2X SSC-T) wash to remove acid, cells were incubated in 2X SSC-T-50% formamide for 20 min at 60°C on a Thermomixer C dry block (Eppendorf 2231001005). Cells

were exchanged into primary hybridization buffer (Hyb1) comprising 2X SSC-T, 50% (vol/vol) formamide, 10% (wt/vol) dextran sulfate, 0.4 µg/µl RNAse A, and ~1 µM extended primary oligos (resuspended dry, unpurified PER reactions). The cell–Hyb1 mixture was distributed into PCR strip tubes at 1e7–1.5e7 cells in 100 µl volumes. The cells were denatured and primary oligos were hybridized to the genome in the PCR strip tubes in a thermocycler using the cycling protocol: 78°C 3 min, 37°C ∞ incubating overnight for more than 18 hr. The next day, cells were rinsed with 60°C 2X SSC-T into 1.5 ml microcentrifuge tubes, followed by two 2X SSC-T buffer exchanges to remove residual Hyb1. Cell pellets were then washed in 1 ml 2X SSC-T at 60°C, followed by two 2-min washes in 2X SSC-T at room temperature. Fully washed cell pellets were exchanged into 1 ml PBS, and then exchanged into 100 nM secondary HRP oligo that map to the PER concatemer sequence on the primary oligo (custom synthesis by Integrated DNA Technologies or Bio-Synthesis Inc) in PBS. Secondary hybridization was performed at 37°C with nutation for 1 hr. Cell pellets underwent three 5 min washes in 1 ml PBS-T at 37°C with nutation. Fully washed cells were incubated in 5 µM desthiobiotin tyramide (Iris Biotech LS-1660) and 1 mM hydrogen peroxide in PBS-T for 5 min at room temperature with nutation. To quench the HRP activity, biotinylated cells were washed twice in 10 mM sodium ascorbate and 10 mM sodium azide in PBS-T for 5 min at room temperature with nutation. Quenched cells were washed with PBS to remove residual sodium azide. After sampling cells for quality control, the cell pellets were stored dry in –80°C until chromatin solubilization and affinity purification.

## On-plate hybridization and biotinylation

EY.T4 oligo hybridizations were performed on cells fixed to glass bottom 6-well plates, with all washes being performed in 1 ml. Cells were rinsed once with fresh PBS, and permeabilized in PBS-0.5% Triton X-100 (Sigma T8787) for 10 min. After a PBS-0.5% Tween20 (PBS-T) (Sigma T2287) wash, permeabilized cells were incubated in 0.1 N HCl for 5 min. After a PBS-T wash to remove acid, cells were incubated in PBS-T-0.5% hydrogen peroxide to block endogenous peroxidases. After a 2X saline sodium citrate-0.1% Tween20 (2X SSC-T) wash to remove acid, cells were incubated in 2X SSC-T-50% formamide for 20 min on a heat block in a 60°C water bath. Cells were exchanged into 500 µl per well of primary hybridization buffer (Hyb1) comprising 2X SSC-T, 50% (vol/vol) formamide, 10% (wt/vol) dextran sulfate, 0.4 µg/µl RNAse A, and ~250 nM extended primary oligos. The cells were denatured on a heat block in a water bath at 78°C for 3 min. The cells were then hybridized with primary oligo at 37°C overnight with nutation for about 24 hr.

The next day, cells were rinsed with 60°C 2X SSC-T four times for 5 min each, followed by two 2-min 2X SSC-T buffer exchanges at room temperature to remove residual Hyb1. Fully washed cells were then washed with 1 ml PBS followed by 100 nM secondary HRP oligo that maps to the concatemer sequence on the primary oligo in PBS. Secondary hybridization was performed at 37°C with nutation for 1 hr. Cells underwent three 5 min washes in PBS-T at 37°C. Fully washed cells were incubated in 5 µM desthiobiotin tyramide (Iris Biotech LS-1660) and 1 mM hydrogen peroxide in PBS-T for 5 min at room temperature. To quench the HRP activity, biotinylated cells were washed twice in 10 mM sodium ascorbate and 10 mM sodium azide in PBS-T for 5 min at room temperature. Quenched cells were washed with PBS to remove residual sodium azide. The cells were stored dry at 4°C until chromatin solubilization and affinity purification.

## Microscopy-based quality control assays for hybridization and biotinylation

We routinely sample cells along the workflow of preparing AP-MS or NGS samples to monitor the specificity of primary oligo hybridization. To assess the quality of primary oligo hybridization for the liquid-phase prepared cells, we sampled roughly 5% of fully washed cells from primary hybridization to a new 1.5 ml tube. Cells were incubated with 400 nM fluorescent oligos in PBS at 37°C for an hour with nutation. Hybridized cells underwent three washes in 1 ml PBS-T at 37°C with nutation to remove unbound fluorescent oligos. Washed cells were immobilized on glass slides with Slowfade Gold Antifade Mountant with DAPI (Thermo Fisher S36938) and coverslips for confocal imaging of FISH signal.

We assessed the quality of biotinylation specificity for all samples entering the proteomics or genomics workflow. For liquid-phase, roughly 5% of fully quenched cells were sampled into a new 1.5 ml tube and incubated with 0.5–1 µg/ml Alexa Fluor 647-streptavidin (Thermo Fisher S32357) in PBS-T, 1% bovine serum albumin at 37°C for 30 min with nutation. Stained cells underwent four

washes in 1 ml PBS-T at 37°C with nutation to remove unbound Alexa Fluor 647-streptavidin conjugate. Washed cells were immobilized on glass slides with Slowfade Gold Antifade Mountant with DAPI and coverslips for confocal imaging of Alexa Fluor 647-streptavidin signals.

For the solid-phase workflow experiments in EY.T4 cells, we assessed primary oligo hybridization and biotinylation specificity simultaneously. A separate 6-well plate was prepared alongside the others that were reserved for quality control with one well per condition. After biotinylation and subsequent washes, this plate was incubated with 0.5–1 µg/ml Alexa Fluor 647-streptavidin (Thermo Fisher S32357) in PBS-T, 1% bovine serum albumin at 37°C for 30 min with nutation. Stained cells underwent four washes in 1 ml PBS-T at 37°C with nutation to remove unbound Alexa Fluor 647-streptavidin conjugate. These cells were then incubated with 400 nM fluorescent oligos in PBS at 37°C for an hour. Hybridized cells underwent three washes in 1 ml PBS-T at 37°C with nutation to remove unbound fluorescent oligos. Washed cells were immobilized on glass slides with Slowfade Gold Antifade Mountant with DAPI and coverslips for confocal imaging of Alexa Fluor 647-streptavidin and FISH signal.

## Confocal microscopy

Confocal imaging was performed using a Yokogawa CSU-W1 SoRa spinning disc confocal device attached to a Nikon ECLIPSE Ti2 microscope. Excitation light was emitted at 30% of maximal intensity from 405, 488, 561, or 640 nm lasers housed inside a Nikon LU-NF laser unit. Laser excitation was delivered via a single-mode optical fiber into the CSU-W1 SoRa unit. Excitation light was directed through a microlens array disk and a SoRa spinning disk containing 50 µm pinholes to the rear aperture of a 100x N.A. 1.49 Apo TIRF oil immersion objective lens by a prism in the base of Ti2. Emission light was collected by the same objective and directed by a prism in the base of Ti2 back into the SoRA unit, where it was relayed by a 1x lens (conventional imaging) or 2.8x lens (super-resolution imaging) through the pinhole disk and then directed to the emission path by a quad-band dichroic mirror (Semrock Di01-T405/488/568/647-13X15X0.5). Emission light was then spectrally filtered by one of four single-bandpass filters (DAPI:Chroma ET455/50M; ATTO488: Chroma ET525/36M; ATTO565:Chroma ET605/50M; Alexa Fluor 647: Chroma ET705/72M) and focused by a 1x relay lens onto an Andor Sona 4.2B-11 camera with a physical pixel size of 11 µm, resulting in an effective resolution of 110 nm (conventional), or 39.3 nm (super-resolution). The Sona was operated in 16-bit mode with rolling shutter readout and exposure times of 70–300 ms.

## FISH-biotinylation co-localization experiment

Fixed cells were split into 5e6 cell aliquots in 1.5 ml microcentrifuge tubes. Primary hybridization and washes were performed similarly to described in the in-solution hybridization and biotinylation of cell pellets with fewer cells. Fully washed cell pellets were exchanged into a secondary co-hybridization buffer containing 30 nM of fluorescent oligos and 100 nM of HRP-oligos in PBS, instead of solely HRP-oligos, for simultaneous hybridization of both species. After washes and biotinylation, the pellets were stained with 0.5–1 µg/ml Alexa Fluor 647-streptavidin. Cells were immobilized on glass slides with Slowfade Gold Antifade Mountant with DAPI and coverslips for confocal imaging of both FISH and Alexa Fluor 647-streptavidin signals.

## FISH-biotinylation co-localization quantification

To quantify the colocalization of signal in the FISH and streptavidin channels, binary masks were generated for each image using a Gaussian blur (scipy.ndimage.gaussian_filter) and power-law transformation. The preprocessed image was then binarized using Otsu's method (skimage.filters. threshold_otsu). Parameter optimization for the sigma value of the Gaussian blur and the exponent values of the power-law transformation were done via grid search. Each parameter set was used to generate a binary mask and scored based on the percentage of pixels labeled as foreground in the mask that exceeded the 90th percentile of pixel intensity values in the input image. Parameters were optimized for each image and channel independently. These binary masks were then used to calculate the labeling efficiency and specificity of the streptavidin signal relative to the ground truth FISH signal and the Jaccard index of both masks. Labeling efficiency is defined as the percentage of pixels labeled as foreground in the FISH mask that were also labeled as foreground in the streptavidin mask. Specificity is defined as the percentage of pixels labeled as foreground in the streptavidin mask that

were also labeled as foreground in the FISH mask. The Jaccard index was calculated by dividing the number of pixels in the intersection of the two masks by the number of pixels in their union.

## Affinity purification and sample preparation for proteomics

For liquid-phase prepared samples, biotinylated cell pellets were removed from –80°C to thaw at room temperature. Each cell pellet was resuspended in roughly 0.9 ml of lysis buffer consisting of 1% SDS and 200 mM EPPS with protease inhibitors (Roche 11836170001). For the EY.T4 solid phase samples, 0.5 ml of lysis buffer was added directly to each well and cells were scrapped and collected in 1.5 ml tubes. The cell mixture was boiled at 95°C for 30 min. The boiled cell mixture was sonicated at 4°C using a Covaris LE-220 focused ultrasonicator with the following protocol: 300 W peak incident power, 50% duty factor, 200 cycles per burst, with a treatment time of 420 s in 1 ml milliTUBEs with AFA fiber (Covaris 520135). The sonicated cell mixture was boiled for a second time at 95°C for 30 min. The boiled lysates were cleared by centrifuging at 21,130 × $g$ for 30 min in an Eppendorf 5424 Microcentrifuge at room temperature. The supernatants were transferred to a fresh 1.5 ml tube. To prevent any remnants of cell debris, the supernatants were cleared for a second time by centrifuging at 21,130 × $g$ for 30 min and the supernatants were transferred to a fresh 1.5 ml tube. The supernatants were stored in –80°C until protein quantification.

The cleared cell lysates were quantified using the Pierce BCA Protein Assay Kit (Thermo Fisher 23225). Pierce Streptavidin Magnetic Beads (Thermo Fisher 88817) were washed using 1% SDS, 200 mM EPPS lysis buffer three times before use. From each labeled cell pellet, 2.17 mg of protein was used to couple with 500 µg of streptavidin beads in a Protein Lo-Bind tube (Eppendorf EP022431081). The lysates were incubated with the bead slurry for 1 hr at room temperature with nutation allowing biotinylated proteins to bind. The coupled beads were collected and separated from the flow-through using a magnetic rack (Sergi Lab Supplies 1005a). After the flow-through was removed, the beads underwent the following washes: 2% SDS with 20 mM EPPS twice, 0.1 M $Na_2CO_3$, 2 M urea, and 1 M KCl with 20 mM EPPS twice. All washes were performed as follows: after immobilizing the beads on a magnetic rack for 5 min, the supernatant was removed, and the beads were resuspended in the new wash buffer and incubated for 5 min with nutation. Finally, the beads were rinsed once with 20 mM EPPS to remove the excess salt.

The washed streptavidin beads were resuspended in 50 µl of 5 mM TCEP, 200 mM EPPS, pH 8.5 for a 20-min on-bead protein reduction. The proteins were alkylated on-bead using 10 mM iodoacetamide for 1 hr in the dark. Then DTT was added to the final concentration of 5 mM to quench the alkylation for 15 min. The beads were rinsed twice with 200 mM EPPS for on-bead digest. For liquid phase samples, assuming 20 µg of eluate protein, 200 ng LysC (Wako) was added to the beads in a 50-µl volume and incubated for 16 hr with vortexing. The next day, 200 ng of trypsin (Promega V5113) was added to the beads and incubated for 6 hr at 37°C at 200 rpm. For solid phase samples, 20 ng of LysC and Trypsin was added, and the samples were processed in the same way. After digestion, the peptide-containing supernatant was collected in a fresh 0.5 ml Protein Lo-Bind tube. The beads were rinsed once with 100 µl 50% acetonitrile, 5% formic acid, and the wash was combined with the peptides. Peptides were desalted via the stop and go extraction (StageTip) (*Rappsilber et al., 2003*) method and dried in a vacuum concentrator.

For label-free analysis of telomere-enriched samples, one sample consisted of HCT-116-Rad21-mAID cells (*Natsume et al., 2016*). For samples intended to be multiplexed, dried, desalted peptides were reconstituted in 4 µl of 200 mM EPPS, pH 8.5. The peptides were labeled using 25 µg of TMTpro 16plex Label Reagents (Thermo Fisher A44520) at 33.3% acetonitrile for 1 hr at room temperature. The labeling reaction was quenched with the addition of 1 µl of 5% hydroxylamine and incubated at room temperature for 15 min. The pooled sample was acidified using formic acid and peptides were desalted using a StageTip cartridge. Peptides were eluted in 70% acetonitrile, 1% formic acid, and dried by vacuum centrifugation.

## Mass spectrometry data acquisition methods and analysis

Samples were resuspended in 5% acetonitrile/2% formic acid prior to being loaded onto an in-house pulled C18 (Thermo Accucore, 2.6 Å, 150 µm) 30 cm column. Peptides were eluted over 180 min gradients running from 96% Buffer A (5% acetonitrile, 0.125% formic acid) and 4% Buffer B (95% acetonitrile, 0.125% formic acid) to 30% Buffer B. Sample eluate was electrosprayed (2700 V) into a

Thermo Scientific Orbitrap Eclipse mass spectrometer for analysis. High field asymmetric waveform ion mobility spectrometry (FAIMS) was set at 'standard' resolution, 4.6 l/min gas flow, and 3 CVs: −40/–60/–80 were used. Briefly, generally for both Label-free and TMTpro analyses, MS1 scans were collected at 120,000 resolving power with a 50 ms max injection time, and the AGC target set to 100%. Peaks from the MS1 scans were filtered by intensity (minimum intensity >5 × 103), charge state ($2 \leq z \leq 6$), and detection of a monoisotopic mass (monoisotopic precursor selection for peptides, MIPS). Dynamic exclusion was used with a duration of 90 s, repeat count of 1, mass tolerance of 10 ppm, and the 'exclude isotopes' option checked. For each MS1, eight data-dependent MS/MS scans were collected. MS/MS scans were conducted in the linear ion trap with the 'rapid' scan rate, 50 ms max injection time, AGC target set to 200%, CID collision energy of 35% with 10 ms activation time (TMTpro) or HCD at 30% collision energy (Label-free), and 0.5 $m/z$ (TMTpro) or 0.7 $m/z$ (Label-free) isolation window. For TMTPro-labeled samples, an MS3 scan was also included in the method. Unless otherwise noted in the methods, the real-time search filter was enabled (*Schweppe et al., 2020*). Using a human fasta downloaded from Uniprot, fixed modifications for the TMTpro mass (+304.207146) were added to n-terminal residues and lysines. Carbamidomethyl (+57.021464) was added for cysteines. Oxidation (+15.9949) was added as a variable modification on methionines. Missed cleavages were set to a maximum of 1. 'TMT mode' was enabled and thresholds of 1 and 0.05 for Xcorr and dCn, respectively, were used as minimums to trigger SPS-MS3 scans. SPS ions were set to 10, and MS3 scans were performed at a resolving power of 50,000, with an HCD collision energy of 45%, AGC of 200%, with a maximum injection time of 200 ms.

Label-free mass spectrometry data for MS1-based quantitation was analyzed with MSFragger (*Kong et al., 2017*) search algorithm searched against a full human protein database for HCT 116 and K562 samples with forward and reverse protein sequences. EY.T4 samples were searched against a full mouse protein database with forward and reverse protein sequences. Fixed modifications included carbamidomethyl (+57.021464) on cysteines. Variable modifications included oxidation (+15.9949) on methionine and formylation (+27.994915) on lysines. Peptides up to 2 missed cleavages were included. Peptide spectral matches and proteins were filtered to a 1% false discovery rate using Percolator (*Käll et al., 2007*).

Multiplexed raw mass spectrometry data was analyzed using the Comet (*Eng et al., 2013*) search algorithm, searched against a full human protein database with forward and reverse protein sequences (Uniprot 10/2020). Precursor monoisotopic peaks were estimated using the Monocle package. Fixed modifications included TMTpro (+304.207146) on N-terminal residues and lysines and carbamidomethyl (+57.021464) on cysteines. Variable modifications included methionine oxidation (+15.9949) and lysine formylation (+27.994915). Peptides with up to two missed cleavages were included. Peptide spectral matches and proteins were filtered to a 1% false discovery rate using the rules of parsimony and protein picking. Protein quantification was done using signal-to-noise estimates of reporter ions. Samples were column normalized for total protein concentration. After filtering for contaminants, we performed a two-sided $t$-test comparing each O-MAP condition using Benjamini–Hochberg adjusted p-values (i.e., $q$-values). Log$_2$ fold changes of the mean of the biological replicates were also calculated for each biological condition. For the GSK treatment experiment, prior to graphing, the data were run through a custom R script using ComBat to correct for batch effects encountered when processing the samples (https://github.com/SchweppeLab/DNA-O-MAP-eLife-HOXAB-GSK126-ComBat; *Herlihy, 2026*). Human Protein Atlas (*Thul et al., 2017*) subcellular locations were downloaded and the 'main location' was assigned to each protein with a supported or enhanced reliability level. SAINT scores and interaction false discovery rates were calculated with the SAINTexpress software (*Choi et al., 2011*; *Teo et al., 2014*). Significant hits were those with a SAINT calculated FDR less than 1% (*Choi et al., 2012*). BioPlex interaction networks were accessed through the online BioPlex Explorer (*Huttlin et al., 2015*). Networks were imaged using Cytoscape 3.10.02 (*Shannon et al., 2003*). Protein complex members were accessed through CORUM (*Ruepp et al., 2008*). Gene set enrichment analysis was performed with clusterProfiler (*Yu et al., 2012*) and fgsea (*Korotkevich et al., 2016*) packages.

## Preparation of soluble chromatin for affinity purification followed by next-generation sequencing

For confirmation of single-copy O-MAP labeling, loop anchor-biotinylated pellets of 10–20 million cells were removed from –80°C to thaw at room temperature. Each cell pellet was resuspended in an SDS lysis buffer consisting of 1% SDS and 200 mM EPPS with protease inhibitors. The cell mixture was sonicated at 4°C using a Covaris LE-220 focused ultrasonicator with the following protocol: 300 W peak incident power, 15% duty factor, 200 cycles per burst, with a treatment time of 20–30 min in 130 µl microTUBEs with AFA fiber (Covaris 520077). After the samples had returned to room temperature, the sheared fixed chromatin was transferred to fresh 1.5 ml Protein Lo-Bind tubes and centrifuged at 21,130 × g for 10 min to pellet cellular debris. The supernatants were transferred to a new set of tubes. The cleared chromatin samples were quantified using the Pierce BCA Protein Assay Kit (Thermo Fisher 23225). Next, 50 µl of sheared chromatin was sampled for reverse crosslinking, DNA extraction, and gel electrophoresis to verify that a significant amount of DNA had been sheared to <700 bp. A sample of 10 µg sheared chromatin was reserved and stored at –20°C as immunoprecipitation input. 200 µg of chromatin was used to couple with 200 µg of streptavidin beads for 1 hr in a Protein Lo-Bind tube at room temperature with nutation. The coupled beads were collected and separated from the flow-through using a magnetic rack. After the flow-through was removed, the beads underwent the following washes:

- 2% SDS with 20 mM EPPS
- High Salt Buffer containing 500 mM NaCl, 1 mM EDTA, 50 mM of HEPES pH 7.5, 0.1% sodium deoxycholate, and 1% Triton X-100
- LiCl Buffer containing 250 mM LiCl, 1 mM EDTA, 10 mM Tris-HCl pH 8.0, and 0.5% of IGEPAL CA-630
- TE Buffer with 10 mM Tris and 1 mM EDTA

The washes were performed as follows: Briefly spin and immobilize the beads on a magnetic rack, pipette out the supernatant as much as possible, resuspend the beads in 0.8 ml of wash buffer, and incubate for 5 min with nutation. The washed beads were resuspended in 300 µl of reverse crosslinking buffer containing 300 mM NaCl, 300 mM Tris-HCl pH 8.0, and 1 mM EDTA. Both the eluate beads and the input chromatin were incubated at 65°C for 16 hr for reverse crosslinking. The next day, 4 µl of 20 mg/ml proteinase K (Roche 3115836001) was added to the eluates and inputs and incubated at 50°C for 2 hr to cleave away proteins. The DNA was isolated from the mixture using phenol-chloroform extraction followed by ethanol precipitation. Before sequencing library generation, the precipitated DNA was further purified using SPRI beads. The purified DNA was used to generate next-generation sequencing libraries using the NEBNext Ultra II DNA Library Prep Kit for Illumina (NEB E7645S) and NEBNext Multiplex Oligos for Illumina Index Primers Set 1 and 3 (NEB E7335S, E7710S) and PCR-amplified for 15 cycles. The sequencing libraries were quantified using the Qubit 4 fluorometer and library sizes were quantified using the D1000 ScreenTape assay (Agilent 5067-5582) on the TapeStation 4200 automated electrophoresis platform.

## DNA sequencing and data analysis

The libraries were mixed and sequenced pair-ended at 50 bp read length on an Illumina NextSeq 2000 sequencer to depths of 14.1–351.8 million reads per eluate sample and 3.14–16.45 millions reads per input sample using the NextSeq 1000/2000 P2 Reagents (100 Cycles) kit (Illumina 20046811). Reads were demultiplexed and adapters were removed using Cutadapt (*Martin, 2011*). Trimmed reads were mapped to the reference genome (GRCh38) using Bowtie2 version 2.5.3 with the parameter -X 1000 keeping reads with a MAPQ ≥ 30 (*Langmead and Salzberg, 2012*). Duplicate reads were removed using Picard 3.1.1 (*Picard, 2026*). Eluate reads were normalized to input reads using DeepTools (*Ramírez et al., 2016*) bamCompare with the following parameters: –binSize 20 –normalizeUsing BPM –smoothLength 60 – extendReads 150. Normalized data were visualized using Coolbox 0.3.9 (*Xu et al., 2021*).

To assess the targeting specificity of DNA O-MAP, O-MAP ChIP sequencing reads were aligned to a reference genome using Bowtie2. Duplicate reads were removed with Picard and Samtools. Enrichment scores were calculated using a bin-based method adapted from the TSA-seq protocol (*Chen et al., 2018*). The genome is divided into 100 kb windows, and the enrichment score for each window

is determined by the following formula: Enrichment Score = (Input reads in bin/Sum of input reads)/ (Pull-down reads in bin/Sum of pull-down reads). To normalize the data and prevent division by zero in bins with no reads, a pseudocount equal to 10% of the total reads was added to each bin. Finally, the calculated enrichment scores were plotted against the log-scaled distance from the target regions.

## Acknowledgements

We would like to thank members of the Shechner, Beliveau, and Schweppe labs for constructive feedback and technical assistance in assembling this work. We would also like to thank Drs. Jay Shendure, Shao-En Ong, Christine Quietsch, Emily Hatch, Gavin Ha, Celeste Berg, Christine Disteche, Andrew Stergachis, and Stanley Fields for helpful discussions of this work. We would like to acknowledge the following sources of support: R35GM137916 (BJB), R35GM150919 (DKS), the W.M. Keck Foundation (BJB, DKS), an Andy Hill CARE Distinguished Researcher Award (DKS), a Damon Runyon Dale Frey Award (BJB), a Cancer Consortium New Investigator Award (funded in part through P30 CA015704, DKS), the Pew Charitable Trusts (DKS), 1R01GM138799-01 and 1R01HL160825-01 (DMS), T32GM007750 (to AFT and EEK), and AHA 902616 (to EEK). Research reported in this publication was supported by the NHLBI under award number T32HL007093 (to CPH). This work was also supported by a Research and Education Training Fund Award (to CH) from the Center for the Multiplex Assessment of Phenotype at UW.

## Additional information

### Competing interests

Evan E Kania, David M Shechner, Brian J Beliveau: has filed a patent application covering aspects of this work (US Patent App. 18/728,937). Devin K Schweppe: has filed a patent application covering aspects of this work (US Patent App. 18/728,937). BJB is also listed as an inventor on patent applications related to the SABER technology related to this work (US Patent 11,492,661; US Patent App. 18/607,269). The other authors declare that no competing interests exist.

### Funding

| Funder | Grant reference number | Author |
| --- | --- | --- |
| National Institute of General Medical Sciences | R35GM137916 | Brian J Beliveau |
| National Institute of General Medical Sciences | R35GM150919 | Devin K Schweppe |
| W.M. Keck Foundation | | Brian J Beliveau Devin K Schweppe |
| Andy Hill CARE Distinguished Researcher Award | | Devin K Schweppe |
| Damon Runyon Dale Frey Award | | Brian J Beliveau |
| Cancer Consortium New Investigator Award | | Devin K Schweppe |
| Pew Charitable Trusts | | Devin K Schweppe |
| National Heart Lung and Blood Institute | T32HL007093 | Conor P Herlihy |
| National Institutes of Health | 1R01GM138799-01 | David M Shechner |
| National Institutes of Health | 1R01HL160825-01 | David M Shechner |

| Funder | Grant reference number | Author |
| --- | --- | --- |
| National Institutes of Health | T32GM007750 | Ashley F Tsue Evan E Kania |
| American Heart Association | AHA 902616 | Evan E Kania |
| CMAP Research and Education Training Fund Award (via NIH RM1 HG010461) | | Chris Hsu |

The funders had no role in study design, data collection, and interpretation, or the decision to submit the work for publication.

## Author contributions

Yuzhen Liu, Christopher D McGann, Conceptualization, Data curation, Formal analysis, Validation, Investigation, Methodology, Writing – original draft, Writing – review and editing; Conor P Herlihy, Conceptualization, Data curation, Formal analysis, Validation, Investigation, Methodology, Writing – original draft, Writing – review and editing, Visualization; Mary Krebs, Qiaoyi Lin, Nicolas J Longhi, Data curation, Formal analysis, Validation, Investigation, Methodology, Writing – review and editing; Thomas A Perkins, Validation, Investigation, Writing – review and editing; Rose Fields, David Z Nwizugbo, Formal analysis, Validation, Investigation, Methodology, Writing – review and editing; Conor K Camplisson, Formal analysis, Validation, Investigation, Writing – review and editing; Chris Hsu, Shayan C Avanessian, Ashley F Tsue, Evan E Kania, Investigation, Methodology, Writing – review and editing; David M Shechner, Supervision, Funding acquisition, Methodology, Writing – review and editing; Brian J Beliveau, Devin K Schweppe, Conceptualization, Data curation, Supervision, Funding acquisition, Writing – original draft, Project administration, Writing – review and editing

## Author ORCIDs

Conor P Herlihy ⓘ https://orcid.org/0000-0001-9818-4204
Brian J Beliveau ⓘ https://orcid.org/0000-0003-1314-3118
Devin K Schweppe ⓘ https://orcid.org/0000-0002-3241-6276

Reviewer #1 (Public review): https://doi.org/10.7554/eLife.102489.3.sa1
Reviewer #2 (Public review): https://doi.org/10.7554/eLife.102489.3.sa2
Reviewer #3 (Public review): https://doi.org/10.7554/eLife.102489.3.sa3
Author response https://doi.org/10.7554/eLife.102489.3.sa4

# Additional files

## Supplementary files

Supplementary file 1. Oligonucleotide probe sequences used in this work.

Supplementary file 2. Proteomic data for the enrichment of telomere probe-associated proteins.

Supplementary file 3. Quantitative proteomic data for the multi-target DNA O-MAP proteomics experiment.

Supplementary file 4. Summary of oligonucleotide probe target information.

Supplementary file 5. Proteomic data for the differential enrichment of *HOXA* and *HOXB* associated proteins.

Supplementary file 6. Quantitative proteomic data for the GSK126 treated *HOXA* proteomics experiment.

Supplementary file 7. Quantitative proteomic data for the GSK126 treated *HOXB* proteomics experiment.

Supplementary file 8. Proteomic data for the differential enrichment of Xi and Xa associated proteins.

MDAR checklist

## 1,2,31231Data availability

The mass spectrometry proteomics data have been deposited to the ProteomeXchange (*Vizcaíno et al., 2014*) Consortium via the MassIVE with the dataset identifier PXD054080. All other primary data associated with the paper (next-generation sequencing, raw microscopy data, and all source data underlying tables and plots in the manuscript) have been deposited in Dryad: https://doi.org/10.5061/dryad.fn2z34v98.

The following dataset was generated:

| Author(s) | Year | Dataset title | Dataset URL | Database and Identifier |
| --- | --- | --- | --- | --- |
| Beliveau BJ, Liu Y, McGann CD, Herlihy CP, Krebs M, Fields R, Camplisson CK, Nwizugbo DZ, Lin Q, Longhi NJ, Hsu C, Avanessian SC, Tsue AF, Kania E, Shechner DM, Schweppe D, Perkins TA | 2026 | Data from: DNA O-MAP uncovers the molecular neighborhoods associated with specific genomic loci | https://doi.org/10.5061/dryad.fn2z34v98 | Dryad Digital Repository, 10.5061/dryad.fn2z34v98 |

The following previously published dataset was used:

| Author(s) | Year | Dataset title | Dataset URL | Database and Identifier |
| --- | --- | --- | --- | --- |
| Schweppe D | 2024 | DNA O-MAP uncovers the molecular neighborhoods associated with specific genomic loci | https://massive.ucsd.edu/ProteoSAFe/dataset.jsp?task=6dbc55a7bba149a2ac8f2e2a61d30b71 | MassIVE, PXD054080 |

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
