## [Editor Report · eLife Assessment]

This study presents an **important** new method for probing the DNA and proteins associated with targeted chromatin domains in cells. The authors present **solid** evidence that the method can map DNA-DNA interactions for individual loci and can detect proteins enriched near repetitive DNA loci or targeted gene clusters. The methodological details of this study will be of particular interest and utility to chromatin biologists.

---

## [Referee Report · Reviewer #1 (Public review)]

The new experiments on the HOX and XIC look strong. A limited (conservative) number of proteins are determined to be enriched at the respective loci. And the number of cells used is a good advancement for these kinds of methods.

Unfortunately, the warnings about mitochondrial to nuclear comparisons and validations do not appear to be taken seriously. It's not that "...there could be non-specific nuclear comparison." There are definitely non-specific enriched proteins. Minimizing false positives is the responsibility of those developing the method and generating the hit lists. I think you saying our probes go to where they are supposed to and label the proteins in that compartment is fine. But that is as far as that should go. Any non-validated protein hits in those comparisons need to be removed. It will contaminate the literature by having all the proteins in 1E, S4D-F, and S5 reported (even though it appears there is no tables reporting the new proteins claimed to be associated with that locus. Why is that?).

I think the line "...we have not made any claims about new proteins at specific loci." is the heart of the issue. What is the point of this method then? Isn't it to identify unknown proteins at a locus of interest? Without that, it's just generating a long list of proteins, where an unknown number of which are likely erroneous, and highlighting the ones you already knew to be there. Along those lines, it is not validation to show proteins that we already knew were at a locus are at the locus. Validation is developing a method to help find new things, then testing those new things to confirm the new method's fidelity.

The comparison of OMAP identified proteins to the several other methods that look at similar regions is not there. A Figure 1F is referred to in the rebuttal but is not in the manuscript. If you mean the Bioplex comparison, that is not the goal. The goal of this analysis to see how much overlap, if any, is being identified across methods. OMAP has so many proteins claimed to be associated with telomeres that are not tested or validated, it would be nice if other methods see similar ones.

Minor points: You have now done label free proteomics. (A) Methodological details are needed. It is not clear if you mean MS1 or DIA based quant. (B) Do you need all the language about how multiplexed proteomics is enabling this methods?

Labeling the all the enriched proteins in the volcano plots would be nice. I don't want to see just the "relevant" ones that support your claims. I want to see all the "new" ones your discovery method is claiming to discover.

---

## [Referee Report · Reviewer #2 (Public review)]

Summary

The authors introduce DNA O-MAP, a method that combines oligo-based in situ hybridization with peroxidase-mediated proximity biotinylation to profile proteins and DNA-DNA interactions linked to targeted genomic regions. In the revised manuscript, they expand the method beyond repetitive elements by profiling non-repetitive gene clusters (HOXA and HOXB), studying inhibitor-induced chromatin remodeling, and differentiating homolog-specific proteomes on both the active and inactive X chromosome. These additions considerably broaden the scope of the work and indicate that DNA O-MAP is currently most effective for analyzing gene-cluster size or domain-level chromatin environments, rather than focusing on individual promoters or cis-regulatory elements.

Strengths

The study demonstrates that DNA O-MAP can be applied to both repetitive domains and non-repetitive genomic regions, including gene clusters spanning 80 kilobases and larger single-copy chromosomal intervals, rather than isolated cis-regulatory elements.

Orthogonal validation using ENCODE ChIP-seq data supports several differentially enriched proteins observed between the HOXA and HOXB gene clusters proteomes.

The ability to detect quantitative changes in local protein environments after chemical perturbation demonstrates the method's sensitivity at the level of extended genomic domains.

Homolog-resolved analysis of the active and inactive X chromosome provides an additional demonstration of biological specificity and technical flexibility at the megabase scale.

The revised manuscript appropriately frames DNA O-MAP as a method for interrogating local domain-level genomic environments, rather than exhaustively defining the protein composition of individual regulatory elements.

Weaknesses

As with all proximity labeling approaches, the effective resolution of DNA O-MAP is constrained by the spatial distance of peroxidase-mediated labeling rather than by genomic distance. Consequently, for gene-cluster-scale targets, enrichment extends beyond the targeted interval into surrounding chromosomal regions, potentially limiting the method's specificity at the level of individual promoters, enhancers, or gene bodies.

Specificity is demonstrated through comparative and internally controlled analyses rather than through a quantitative estimate of false discovery rate for locus specificity. Readers should therefore interpret individual protein enrichments as indicative of local chromatin environments rather than definitive evidence of direct binding to a specific regulatory element.

Orthogonal validation is necessarily selective and hypothesis-driven. A broader validation would be required before newly enriched proteins can be interpreted as bona fide region-resident factors.

Comparisons to prior locus-proteomics methods remain indirect and should be interpreted primarily in terms of demonstrated feasibility, scalability, and reduced cell-number requirements rather than absolute performance or resolution.

---

## [Referee Report · Reviewer #3 (Public review)]

Significance of the Findings:

The study by Liu et al. presents a novel method, DNA-O-MAP, which combines locus-specific hybridisation with proximity biotinylation to isolate specific genomic regions and their associated proteins. The potential significance of this approach lies in its purported ability to target genomic loci with heightened specificity by enabling extensive washing prior to the biotinylation reaction, theoretically improving the signal-to-noise ratio when compared with other methods such as dCas9-based techniques. Should the method prove successful, it could represent a notable advancement in the field of chromatin biology, particularly in establishing the proteomes of individual chromatin regions-an extremely challenging objective that has not yet been comprehensively addressed by existing methodologies.

Strength of the Evidence:

The evidence presented by the authors is somewhat mixed, and the robustness of the findings appears to be preliminary at this stage. While certain data indicate that DNA-O-MAP may function effectively for repetitive DNA regions, a number of the claims made in the manuscript are either unsupported or require further substantiation. There are significant concerns about the resolution of the method, with substantial biotinylation signals extending well beyond the intended target regions (megabases around the target), suggesting a lack of specificity and poor resolution, particularly for smaller loci. Furthermore, comparisons with previous techniques are unfounded since the authors have not provided direct comparisons with the same mass spectrometry (MS) equipment and protocols. Additionally, although the authors assert an advantage in multiplexing, this claim appears overstated, as previous methods could achieve similar outcomes through TMT multiplexing. Therefore, while the method has potential, the evidence requires more rigorous support, comprehensive benchmarking, and further experimental validation to demonstrate the claimed improvements in specificity and practical applicability.

---

## [Author Response]

The following is the authors’ response to the original reviews.

**Public Reviews:**

**Reviewer #1 (Public review):**
Summary:The authors describe a method to probe both the proteins associated with genomic elements in cells, as well as 3D contacts between sites in chromatin. The approach is interesting and promising, and it is great to see a proximity labeling method like this that can make both proteins and 3D contacts. It utilizes DNA oligomers, which will likely make it a widely adopted method. However, the manuscript over-interprets its successes, which are likely due to the limited appropriate controls, and of any validation experiments. I think the study requires better proteomic controls, and some validation experiments of the "new" proteins and 3D contacts described. In addition, toning down the claims made in the paper would assist those looking to implement one of the various available proximity labeling methods and would make this manuscript more reliable to non-experts.Strengths:(1) The mapping of 3D contacts for 20 kb regions using proximity labeling is beautiful.(2) The use of in situ hybridization will probably improve background and specificity.(3) The use of fixed cells should prove enabling and is a strong alternative to similar, living cell methods.Weaknesses:(1) A major drawback to the experimental approach of this study is the "multiplexed comparisons". Using the mtDNA as a comparator is not a great comparison - there is no reason to think the telomeres/centrosomes would look like mtDNA as a whole. The mito proteome is much less complex. It is going to provide a large number of false positives. The centromere/telomere comparison is ok, if one is interested in what's different between those two repetitive elements.

We appreciate the reviewers' point here. In fact we selected the mitochondrial DNA as a target for just the reason that the reviewer notes. mtDNA should be spatially distinct from the nuclear targets and allow us to determine if we were in fact seeing spatially distinct proteins at the interorganelle (mtDNA vs. telomeres/centrosomes) and intraorganelle (telomeres vs centromeres) levels.

But the more realistic use case of this method would be "what is at a specific genomic element"? A purely nuclear-localized control would be needed for that. Or a genomic element that has nothing interesting at it (I do not know of one).

We have now added two studies in Figure 4 and Figure 5 detailing the use of OMAP to investigate specific genomic elements. In this case the Hox clusters (*HOXA* and *HOXB*) and haplotype-specific analysis of X-chromosome inactivation centers in female murine (EY.T4) cells. The controls in these cases are more specific, in line with those suggested by the reviewer as we (1) compare *HOXA* and *HOXB* with or without EZH2 inhibition using the same sets of probes and (2) specifically compare the region surrounding the XIC in female cells for the inactive and active X chromosomes.

You can see this in the label-free work: non-specific, nuclear GO terms are enriched likely due to the random plus non-random labeling in the nucleus. What would a Telo vs general nucleus GSEA look like? (GSEA should be used for quantitative data, no GO). That would provide some specificity. Figures 2G and S4A are encouraging, but (a) these proteins are largely sequestered in their respective locations, and (b) no validation by an orthogonal method like ChIP or Cut and Run/Tag is used.

We performed GSEA on the enrichment scores for the label-free proteomics data from the SAINT output in Figure 1D and that several of these proteins (e.g., those highlighted in Figure 2A: TERF1, CENPN, TOM70) have already been extensively validated to co-localize to these locations.

To the reviewers request for additional validation, we analyzed ChIP-seq data for several proteins to determine if they were enriched surrounding specific loci. In the case of the HoxA/B analysis, we found that HDAC3 and TCF12 were enriched at *HOXB* compared to *HOXA*, and SMARCB1 and ZC3H13 were enriched at *HOXA* compared to *HOXB* (Figure 4C). HDAC3 and TCF12 ChIP data confirmed increased peak calls at *HOXB* and SMARCB1 and ZC3H13 ChIP data confirmed increased peak calls at HOXA for these four selected proteins (Figure 4D).

You can also see this in the enormous number of "enriched" proteins in the supplemental volcano plots. The hypothesis-supporting ones are labeled, but do the authors really believe all of those proteins are specific to the loci being looked at? Maybe compared to mitochondria, but it's hard to believe there are not a lot of false positives in those blue clouds. I believe the authors are more seeing mito vs nucleus + Telo than the stated comparison. For example, if you have no labeling in the nucleus in the control (Figures 1C and 2C) you cannot separate background labeling from specific labeling. Same with mito vs. nuc+Telo. It is not the proper control to say what is specifically at the Telo.

We agree with the reviewer that compared to mitochondrial targeting, there could be non-specific nuclear comparisons. We note again though that we purposefully stayed away from using the word “specifically” when describing the proteomics work developed here. The reason being that we are not atlasing a large number of targets to define specificity. Instead, we highlight in Figure 2 that we did observe differences in proteins associating with telomeres and mitochondrial DNA. That may be non-specific, and in fact, this is also why we decided to include two nuclear targets to determine what might be specifically enriched. Thus, we compared centromeric and telomeric protein enrichment as determined by OMAP and observed consistent differential enrichment of shelterin proteins at telomeres (Figure 2I) and CENP-A complex members at centromeres (Figure 2J). We could have done the relative comparisons to no-oligo controls, analogous to how CASPEX compared targeted analyses to no-sgRNA controls (PMID: 29735997). However, we found that the mitochondrial targeted samples were generally better as a comparator because (1) we have clear means to validate differences and (2) the local environment around DNA is being labeled.

I would like to see a Telo vs nuclear control and a Centromere vs nuc control. One could then subtract the background from both experiments, then contrast Telo vs Cent for a proper, rigorous comparison. However, I realize that is a lot of work, so rewriting the manuscript to better and more accurately reflect what was accomplished here, and its limitations, would suffice.

Assuming the nuclear control was the same, It is unclear how this ratio-of-ratios ([Telo/Ctrl]/[Cent/ctrl]) experiment would be inherently different from the direct comparison between Telo and Centromere. Again, assuming the backgrounds are derived from the same cellular samples. More than likely adding the extra ratios could increase the artifactual variance in the estimates, reducing the power of the comparisons as has been seen in proteomics data using ratio-of-ratio comparisons in the past (Super-SILAC).

(2) A second major drawback is the lack of validation experiments. References to literature are helpful but do not make up for the lack of validation of a new method claiming new protein-DNA or DNA-DNA interactions. At least a handful of newly described proximal proteins need to be validated by an orthogonal method, like ChIP qPCR, other genomic methods, or gel shifts if they are likely to directly bind DNA. It is ok to have false positives in a challenging assay like this. But it needs to be well and clearly estimated and communicated.

We appreciate the reviewers' point here. To be clear, we have not made any claims about new proteins at specific loci. Instead we validated that known telomeric and centromeric associating proteins were consistently enriched by DNA OMAP (Figure 2). We also want to emphasize that while valuable, the current paper is not an atlasing paper to define the full and specific proteomes of two genomic loci. We instead show how this method can be used to observe quantitative differences in proteins enriched at certain loci (*HOXA/B* work, Figure 4) and even between haplotypes (Xi/Xa work, Figure 5).

(3) The mapping of 3D contacts for 20 kb regions is beautiful. Some added discussion on this method's benefits over HiC-variants would be welcomed.

We appreciate the reviewers' point here and have added the following text to the discussion: “Additionally, we show that this method is also able to detect DNA-DNA contacts through biotinylation of loop anchors. Our approach functions similarly to 4C[86]. However, our approach of biotin labeling of contacts does not rely on pairwise ligation events. Thus, detection of contacts through DNA O-MAP will vary in the sampling of DNA-DNA contacts in comparison.”

(4) The study claims this method circumvents the need for transfectable cells. However, the authors go on to describe how they needed tons of cells, now in solution, to get it to work. The intro should be more in line with what was actually accomplished.

We took the reviewers point and have worked to scale down the DNA OMAP experiments while revising this manuscript. As noted in Figure 5, we have been able to scale this work down to work on plates with ~10x fewer cells than with our initial experiments. This is on top of the initial DNA OMAP work in Figure 1 and 2, as well as our additional work in Figure 4, where we are using 30-60 million cells in solutions which is still 10x less material than previous work (PMID: 29735997). Thus, the newest DNA OMAP platform uses ~100x fewer cells than previous work.

(5) Comments like "Compared to other repetitive elements in the human genome...." appear to circumvent the fact that this method is still (apparently) largely limited to repetitive elements. Other than Glopro, which did analyze non-repetitive promoter elements, most comparable methods looked at telomeres. So, this isn't quite the advancement you are implying. Plus, the overlap with telomeric proteins and other studies should be addressed. However, that will be challenging due to the controls used here, discussed above.

As noted above, we have added Figures 4 and 5 to address the reviewer concerns by targeting multiple non-repetitive loci (*HOXA* and *HOXB* clusters and a 4.5Mb region straddling X-inactivation center on both the active and inactive X homolog). Targeting the regions around the X-inactivation center shows the potential to perform haplotype-resolved proteome analysis of chromatin interactors.

For the telomeric protein overlap, we tried to do this specifically in Figure 1F, we agree with the reviewer that the controls used dramatically change the proteins considered enriched. The goal of the network analysis was to show (1) that we identify proteins previously observed in telomere proteomic datasets and (2) that we gain a more complete view of proteins based on capturing more known interacting proteins than many previous methods as was noted for the RNA OMAP platform (PMID: 39468212). For example, we observed enrichment of PRPF40A in the telomeric DNA OMAP data. From the Bioplex interactome, PRPF40A was observed to interact with TERF2IP and TERF2, suggesting that through these interactions PRPF40A may colocalize at telomeres. Similarly, we observed enrichment of SF3A1, SF3B1, and SF3B2. The SF3 proteins are known regulators of telomere maintenance (PMID: 27818134), but have not previously been observed in telomeric proteomics datasets, except now in DNA OMAP.

We have added the following text to the Results to clarify these points:

“To benchmark DNA O-MAP, we compared the full set of telomeric proteins to proteins observed in five established telomeric datasets (PICh, C-BERST, CAPLOCUS, CAPTURE, BioID)12,14,16,35,36 (Figure 1F). DNA O-MAP captured both previously observed telomeric interacting proteins (shelterins) as well as telomere associated proteins (ribonucleoproteins). We identified multiple heterogeneous nuclear ribonucleoproteins (hnRNPs) previously annotated as telomere-associated, including HNRNPA1 and HNRNPU. HNRNPA1 has been demonstrated to displace replication protein A (RPA) and directly interact with single-stranded telomeric DNA to regulate telomerase activity37–39. HNRNPU belongs to the telomerase-associated proteome40 where it binds the telomeric G-quadruplex to prevent RPA from recognizing chromosome ends41. We mapped DNA O-MAP enriched telomeric proteins to the BioPlex protein interactome and observed that in addition to capturing proteins from previously observed telomeric datasets (Figure 1F), DNA O-MAP enriched for interactors of previously observed telomeric proteins. Previous data found RBM17 and SNRPA1 at telomeres, and in BioPlex these proteins interact with three SF3 proteins (SF3A1, SF3B1, SF3B2). Though they were not identified in previous telomeric proteome datasets, all three of these SF3 proteins were enriched in the DNA O-MAP telomeric data. Furthermore, through interactions with G-quadruplex binding factors, these SF3 proteins are regulators of telomere maintenance (PMID: 27818134). Taken together, this data supports the effectiveness of DNA O-MAP for sensitively and selectively isolating loci-specific proteomes.”

**Reviewer #2 (Public review):**
SummaryLiu and MacGann et al. introduce the method DNA O-MAP that uses oligo-based ISH probes to recruit horseradish peroxidase for targeted proximity biotinylation at specific DNA loci. The method's specificity was tested by profiling the proteomic composition at repetitive DNA loci such as telomeres and pericentromeric alpha satellite repeats. In addition, the authors provide proof-of-principle for the capture and mapping of contact frequencies between individual DNA loop anchors.StrengthsIdentifying locus-specific proteomes still represents a major technical challenge and remains an outstanding issue (1). Theoretically, this method could benefit from the specificity of ISH probes and be applied to identify proteomes at non-repetitive DNA loci. This method also requires significantly fewer cells than other ISH- or dCas9-based locus-enrichment methods. Another potential advantage to be tested is the lack of cell line engineering that allows its application to primary cell lines or tissue.

We thank the reviewers for their comments and note that we have followed up on the idea of targeting non-repetitive DNA loci (*HOXA* and *HOXB* clusters and a 4.5Mb section of the X chromosome on each homolog) in the revised manuscript (Figures 4 and 5).

WeaknessesThe authors indicate that DNA O-MAP is superior to other methods for identifying locus-specific proteomes. Still, no proof exists that this method could uncover proteomes at non-repetitive DNA loci. Also, there is very little validation of novel factors to confirm the superiority of the technique regarding specificity.

Our primary claim for DNA OMAP is that it requires orders of magnitude fewer cells than previous studies. Based on comments along these lines from both reviewers, we performed DNA OMAP targeting non-repetitive DNA loci (*HOXA* and *HOXB* clusters and a 4.5Mb section of the X chromosome on each homolog) in the revised manuscript (Figure 4 and 5). For the X chromosome targeting, we used ~3 million cells per condition with methods that we optimized during revision. When targeting *HOXA* and *HOXA*, we were able to identify HDAC3 and TCF12 enrichment at *HOXB* compared to *HOXA* as well as ZC3H13 and SMARB1 enrichment at *HOXA* compared to *HOXB*, which is consistent with ChIP-seq reads from ENCODE for these proteins (Figure 4C, D). Both the *HOX*and X chromosome work help to address limitations noted in the Gauchier et al. paper the reviewer notes as both show progress towards overcoming “the major signal-to-noise ratio problem will need to be addressed before they can fully describe the specific composition of single-copy loci”.

The authors first tested their method's specificity at repetitive telomeric regions, and like other approaches, expected low-abundant telomere-specific proteins were absent (for example, all subunits of the telomerase holoenzyme complex). Detecting known proteins while identifying noncanonical and unexpected protein factors with high confidence could indicate that DNA O-MAP does not fully capture biologically crucial proteins due to insufficient enrichment of locus-specific factors. The newly identified proteins in Figure 1E might still be relevant, but independent validation is missing entirely. In my opinion, the current data cannot be interpreted as successfully describing local protein composition.

We analyzed ChIP-seq reads for our *HOXA* and *HOXB* (Figure 4C,D) which recapitulate our findings for four of our differentially enriched proteins. We also note that with the addition of the nonrepetitive loci (Figures 4 and 5), we have performed DNA OMAP on seven different targets (telomeres, pericentromeres, mitoDNA, *HOXA*, *HOXB*, Xi, and Xa) and identified expected targets at each of these. The consistency of these data, which mirrors the consistency of the RNA implementation of OMAP (PMID: 39468212), reinforces that we can successfully enrich local proteomes at genomic loci.

Finally, the authors could have discussed the limitations of DNA O-MAP and made a fair comparison to other existing methods (2-5). Unlike targeted proximity biotinylation methods, DNA O-MAP requires paraformaldehyde crosslinking, which has several disadvantages. For instance, transient protein-protein interactions may not be efficiently retained on crosslinked chromatin. Similarly, some proteins may not be crosslinked by formaldehyde and thus will be lost during preparation (6).

Based on this critique we have gone back through the manuscript to improve the fairness of our comparisons and expanded the limitations in our discussion section.

To the point about fixation, Schmiedeberg et al., which the reviewer references, does describe crosslinking requiring longer interactions (~5 s). Yet, as featured in reviews, many additional studies have found that “it has been possible to perform ChIP on transcription factors whose interactions with chromatin are known from imaging studies to be highly transient” (Review PMID: 26354429). We note similar results in proteomics analysis in Subbotin and Chait that state that the linkage of lysine-based fixatives like formaldehyde and “glutaraldehyde to reactive amines within the cellular milieu were sufficient to preserve even labile and transient interactions (PMID: 25172955).

(1) Gauchier M, van Mierlo G, Vermeulen M, Dejardin J. Purification and enrichment of specific chromatin loci. Nat Methods. 2020;17(4):380-9.(2) Dejardin J, Kingston RE. Purification of proteins associated with specific genomic Loci. Cell. 2009;136(1):175-86.(3) Liu X, Zhang Y, Chen Y, Li M, Zhou F, Li K, et al. In Situ Capture of Chromatin Interactions by Biotinylated dCas9. Cell. 2017;170(5):1028-43 e19.(4) Villasenor R, Pfaendler R, Ambrosi C, Butz S, Giuliani S, Bryan E, et al. ChromID identifies the protein interactome at chromatin marks. Nat Biotechnol. 2020;38(6):728-36.(5) Santos-Barriopedro I, van Mierlo G, Vermeulen M. Off-the-shelf proximity biotinylation for interaction proteomics. Nat Commun. 2021;12(1):5015.(6) Schmiedeberg L, Skene P, Deaton A, Bird A. A temporal threshold for formaldehyde crosslinking and fixation. PLoS One. 2009;4(2):e4636.
**Reviewer #3 (Public review):**
Significance of the Findings:The study by Liu et al. presents a novel method, DNA-O-MAP, which combines locus-specific hybridisation with proximity biotinylation to isolate specific genomic regions and their associated proteins. The potential significance of this approach lies in its purported ability to target genomic loci with heightened specificity by enabling extensive washing prior to the biotinylation reaction, theoretically improving the signal-to-noise ratio when compared with other methods such as dCas9-based techniques. Should the method prove successful, it could represent a notable advancement in the field of chromatin biology, particularly in establishing the proteomes of individual chromatin regions - an extremely challenging objective that has not yet been comprehensively addressed by existing methodologies.Strength of the Evidence:The evidence presented by the authors is somewhat mixed, and the robustness of the findings appears to be preliminary at this stage. While certain data indicate that DNA-O-MAP may function effectively for repetitive DNA regions, a number of the claims made in the manuscript are either unsupported or require further substantiation. There are significant concerns about the resolution of the method, with substantial biotinylation signals extending well beyond the intended target regions (megabases around the target), suggesting a lack of specificity and poor resolution, particularly for smaller loci.

We thank the reviewers for their comments and note that we have followed up on the idea of targeting non-repetitive DNA loci (HOX clusters and part of the X chromosome) in the revised manuscript (**Figures 4 and 5**).

Furthermore, comparisons with previous techniques are unfounded since the authors have not provided direct comparisons with the same mass spectrometry (MS) equipment and protocols. Additionally, although the authors assert an advantage in multiplexing, this claim appears overstated, as previous methods could achieve similar outcomes through TMT multiplexing. Therefore, while the method has potential, the evidence requires more rigorous support, comprehensive benchmarking, and further experimental validation to demonstrate the claimed improvements in specificity and practical applicability.

We have made the comparisons as best as possible. In fact, we found it difficult to find examples of recent implementations of many of these methods. Purchasing the exact mass spectrometers or performing every version of chromatin proteomics would be well beyond the scope of this work. On the other hand, OMAP has already generated data for three manuscripts. We are making the claim that using the instrumentation and methods available to us, we were able to reduce the number of cells required to analyze a given genomic loci. We then applied TMT multiplexing to further improve the throughput and perform replicate analyses. To fully validate that one protein exists at one loci and no other would require exhaustive atlasing of protein-genomic interactions which would be well beyond the scope of this single paper. Similarly, ChIP for every target identified to assess an empirical FDR would be well beyond the scope of this work.

**Recommendations for the authors:**

**Reviewing Editor Comments:**
In summary, all three reviewers raised major concerns about the limitations of the method, many of which could be resolved by more precise and transparent language about these limitations. If you choose to resubmit a revised version, you should address questions like: What scale does "individual locus" refer to? At what scale can the method map protein-DNA interactions at individual targeted loci, rather than large repetitive domains? What is the estimated false discovery rate for a set of enriched proteins? The eLife assessment for this version of the manuscript is based on reviewer concerns. Note that this assessment can be updated after receiving a response to reviewer comments.
**Reviewer #1 (Recommendations for the authors):**
(1)The first couple of paragraphs make it sound like your method would exclusively benefit from sample multiplexing with MS-based proteomics. That is a bit misleading. The other stated methods use TMT. They don't use it to compare very different genomic (or compartmental) regions, but there is no reason cberst, glopro or CasID could not.

A good point and we have updated the manuscript to reflect this. While previous methods generally did not use TMT, they could be adapted to do so and, similar to OMAP, improved by the use of more replicates in their analyses.

(2) Please make the colors in 1F for the dataset overlap easier to read. 2 and 4+ are too similar.

We appreciate the comment on making the colors easier to discern. Along these lines we’ve changed the color of “2” to make it easier to distinguish from “4+”.

(3) Label as many dots as legible in your volcano plots.

We’ve labeled a number of proteins that are relevant to the discussion in this paper as well as some additional proteins. We feel that additional labeling would detract from the points that we are trying to make in individual figure panels about groups of proteins, rather than general remodeling of all proteins.

(4) Figure 2E needs a divergent color scheme since it crosses 0. And is it scaled, log-transformed, or both? And compared to what then?

Figure 2E (heatmap) is z-scaled relative protein abundance measurements based on TMTpro reporter ion signal to noise (“s/n”). We have added additional information to the legend to highlight the information that the reviewer points out here. For the color, we are unsure of what is being asked for, as above 0 is red and below 0 is blue.

(5) Unclear what you are implying with "...only 1-2 biological replicates." I would omit or clarify.

Fair point, we have updated the manuscript to omit this section to simplify the introduction.

(6) H2O2 and biotin phenols might be toxic to living organisms. But so is 4% PFA and ISH. I realize you are trying to justify your new approach but you don't need to do it with exaggerated contrasts. This O-MAP is a great approach and probably more likely for people to adopt it because it's DNA ISH based. Plus, with the clinking, you are likely not displacing proteins via Cas9 landing.

We appreciate the reviewer’s comments about adoption and lack of protein displacement. We’ve scaled back on the claims and added more about limitations owing to crosslinking and ISH.

(7) How much genome does the Cent regions take up? You state 500 kb for Telos.

In the text we delineate how large of a region the PanAlpha probes target “The genome-wide binding profile of the pan-alpha probe closely overlaps with centromeres (Figure S1) and covers approximately 35 Mb of the genome according to *in silico* predictions.” Additionally, we’ve added Table S4 to summarize target locus sizes for all of the included targets.

(8) You seem to be underestimating the lysine labeling. Is that after TMT labeling and analysis? If so, you're already ignoring what couldn't be seen. I don't think it's that important but you included it, so please describe clearly why it's an issue and how much of an issue it is. How does that relate to lit values? And it's not just TMTpro, it's any lysine labeler.

We appreciate the reviewers point about specifying the reasoning and the lack of clarity around overall lysine labeling. That 1.38% is the number of peptides with remainder modifications due to formaldehyde crosslinking. For overall acylation of lysines with TMT labels, we generally expect (and achieve) >97% labeling of lysines with TMT reagents as the Kuster and Carr labs nicely demonstrated across a range of labeling conditions (PMID: 30967486).

Decrosslinking is a critical step generally for proteomics workflows on fixed or FFPE tissues and thus we sought to explore whether we could achieve sufficiently low residual lysine alkylation to enable protein quantitation by TMTpro reagents (or any lysine labeler, as the reviewer notes). For TMTpro-based methods on peptides, this is less of a concern generally as protease cleavage frees new primary amines at the N-termini of peptides which can be labeled for quantitation. But in part since we are describing a proteomics method on fixed tissues we wanted to share these data and the potential inclusion of residual fixation modifications for readers to potentially take into consideration when performing this method.

**Reviewer #3 (Recommendations for the authors):**
Liu et al. describe an original locus labelling approach that enables the isolation of specific genomic regions and their associated proteins. I have mixed views on this work, which, in my opinion, remains preliminary at this stage. Establishing the proteome of a single chromatin region is one of the most complex challenges in chromatin biology, as extensively discussed in Gauchier et al. (2020). Any breakthrough towards this goal is of significant interest to the community, making this manuscript potentially compelling. Indeed, some data suggest that the method works for repetitive DNA to some extent. However, much of the data is not very convincing, and in the case of small DNA targets, it argues against the use of DNA-O-MAP.In contrast to existing methods, DNA-O-MAP combines locus-specific hybridisation in situ (using affordable oligonucleotides) with proximity biotinylation. A major advantage of this strategy over other locus-specific biotinylation methods is the possibility of extensively washing excess or non-specifically hybridised probes before the biotinylation reaction, theoretically limiting biotinylation to the target region and thus significantly enhancing the signal-to-noise ratio. Other methods involving proximity biotinylation, such as targeted dCas9, do not have this capacity, meaning biotinylation occurs not only at the locus where a small fraction of dCas9 molecules is targeted but also around non-bound dCas9 molecules (representing the vast majority of dCas9 expressed in a given cell). This aspect potentially represents an interesting advance.

We thank the reviewer for their thoughts and critiques, which we hope have in part relieved concerns pertaining to limitation on repetitive elements. To the latter points, we confirmed this with new specificity analysis that showed labeling to be highly specific to a given probe locus (Figure S3).

Below, I outline the significant issues:The manuscript implies that DNA-O-MAP has better sensitivity than earlier techniques like CAPTURE, GLOPRO, or PICh. The authors state that PICh uses one trillion cells (which I doubt is accurate), and other methods require 300 million cells, whereas DNA-O-MAP uses only 60 million cells, suggesting the latter is more feasible. However, these earlier experiments were conducted almost 15 and 6 years ago, when mass spectrometry (MS) sensitivity was considerably lower than that of current instruments. The authors cannot know whether the proteome obtained by previous methods using 60 million cells, but analysed with current MS technology, would yield results inferior to those of DNA-O-MAP. Unless the authors directly compare these methods using the same number of cells and identical MS setups, I find their argument unjustified and misleading.

Based on the instrumentation listed, we actually do have a good idea of how sensitivity changes may have affected identifications and overall sensitivity. For example, the CASPEX data was collected on an Orbitrap Fusion Lumos, while our data was collected on an Orbitrap Fusion Eclipse. From our work characterizing these two instruments during the Eclipse development (PMID: 32250601), we do actually know that the ion optics improvements boosted sensitivity of the Eclipse used in our work compared to the Lumos by ~50%, meaning if GLOPRO was run on an Eclipse it would still require >200 million cells per replicate for input.

It is suggested that DNA-O-MAP is capable of 'multiplexing', whereas previous methods are not. This statement is also misleading. As I understand it, the targeted regions do not originate from a common pool of cells. Instead, TMT multiplexing only occurs after each group of cells has been independently labelled (Telo, Centro, Mito, control). Therefore, previous methods could also perform multiplexing with TMT. Moreover, it is unclear how each proteome was compared: one would expect many more proteins from centromeres than from telomeres (I am unsure about the number of mitochondria in these cells) since these regions are significantly larger than telomeres (possibly 10 to 100 times larger?). Have the authors attempted to normalise their proteomics data to the size (concatenated) of each target? This is particularly relevant when comparing histone enrichment at chromatin regions of differing sizes.

We agree with the reviewers that this was overstated. In fact the GLOPRO paper notes that they performed a MYC analysis with a previous generation of TMT that could multiplex 10 samples. We have amended the manuscript to be more specific in those contexts. As stated in the methods section, “Samples were column normalized for total protein concentration”, to account for the amount of protein and size of the different targets.

Figure 1C shows streptavidin dots resembling telomeres. To substantiate this claim, simultaneous immunofluorescence with a telomere-specific protein (e.g., TRF1 or TRF2) is required. It is currently unknown whether all or only a subset of telomeres are targeted by DNA-O-MAP, and it is also unclear if some streptavidin foci are non-telomeric. Quantification is needed to indicate the reproducibility of the labelling (the same comment applies to the centromere probes later in the manuscript; an immunofluorescence assay with CENPB would be informative, alongside quantifications).

We understand the reviewer’s concern about specificity and reproducibility of DNA-O-MAP. To address this we have added analysis showing the efficiency and specificity of our FISH and biotin labeling for Telomere, PanAlpha, and Mitochondria targeting oligos (Figure S3). We found that biotin deposition was highly specific to the intended targets with an average across the three probes of 98% specificity.

Perhaps more importantly, the authors suggest that it may be possible to enrich proteins that are not necessarily present at the target locus but are instead in spatial proximity (e.g., RNA polymerase I subunits enriched upon centromere targeting). Does this not undermine the purpose of retrieving locus-specific proteomes?

The goal of DNA OMAP is to identify a local neighborhood of proteins around a specific genomic loci, similar to GLOPRO. As we note in the work presented in Figure 4 and 5 now, these neighborhoods are inherently interesting for comparison of quantitative changes that occur around a genomic locus.

Possibly related to the previous issue, when DNA-O-MAP is used to assess DNA-DNA interactions, probes covering regions of 20-25 kb are employed. Therefore, one would expect these regions to be significantly biotinylated compared to flanking regions. However, Genome Browser screenshots indicate extensive biotinylation signals spanning several megabases around the 20-25 kb targets. If the method were highly resolutive, the target region would be primarily enriched, with possibly discrete lower enrichment at distant interacting regions. The lack of discrete enrichment suggests poor resolution, likely due to the likely large scale of proximity biotinylation. This compromises the effectiveness of DNA-O-MAP, especially if it is intended to target small loci with complex sequences. Could the authors quantify the absolute number of reads from the target region compared to those from elsewhere in the genome (both megabases around the locus and other chromosomes, where many co-enriched regions seem to exist)? This would provide insights into both enrichment and specificity.

Thanks for this suggestion, we have included a new Figure S8 to look at normalized read depth as a function of distance from the genomic target. The resolution of DNA OMAP, like all peroxidase mediated proximity labeling methods, is not dependent on the sequence length of the DNA region, but the 30-40nm of physical space around the HRP molecule that is targeted to the genomic loci.

Minor Issues:(1) Page 3, second paragraph: It is unclear why probes producing a visible signal in situ necessarily translates to their ability to retrieve a specific proteome.

We have revised the manuscript to de-emphasize the visible signal aspect of probe targeting and re-emphasize our initial point that the number of probes needed to properly target unique regions makes the use of locked nucleic acid probes cost-prohibitive. The basic point though, we and others previously showed with RNA OMAP (PMID: 39468212) and Apex/proximity labeling strategies, the ability to deposit biotin and visualize generally directly translates to recovery of proximally labeled proteins (PMID: 26866790).

(2) Page 3, last paragraph: "to reach a higher degree of enrichment...": Has it been demonstrated that direct protein biotinylation provides higher enrichment of relevant proteins? Certainly, there is higher enrichment of proteins, but whether they are relevant is another matter.

Our point here was that the methods using direct protein biotinylation have higher levels of enrichment and thus require less cells than the previously mentioned PICh method, which is why we wrote the following: “In the case of GLoPro, APEX-based proximity labeling enhanced protein detection sensitivity, reducing the input required for each replicate analysis to ~300 million cells—a 10-fold reduction in cell input compared to PICh which used 3 billion cells.”

Regarding if these proteins are relevant or not, we show enrichment of known proteins that are critical to the function of their occupied genomic region at telomeres and centromeres. Additionally, we’ve made added quantitative comparisons to assess relevance in our analysis of Hox and our targeted region of the X chromosome through comparisons to ChIP data at these regions. The improved enrichment that we’ve established in our initial submission as well as in the updated version also means that we can further scale down the number of cells required.

(3) Figure 2B is misleading; it appears as though all three regions are targeted in the same cell, suggesting true multiplexing, which, I believe, is not the case.

To avoid any potential confusion about how the samples were derived we’ve updated this figure panel to show three separate cells, each with a different region being targeted.

(3) If I understand correctly, the 'no probe' control should primarily retrieve endogenously biotinylated proteins (carboxylases), which are mainly found in mitochondria. Why does the Pearson clustering in Supplementary Figure 2 not place this control proteome closer to the mitochondrial proteome?

Under the assumption that the ~10 carboxylases are biotinylated at the same levels in all cells, yet the proportion of these carboxylases compared to all enriched proteins for a given target is markedly reduced. Thus, as a proportion of the enriched proteome we note in Figure S4 that mitochondrial DNA OMAP enriches proteins besides the carboxylases. We believe this explains why the ‘no probe’ sample can be clearly separated along PC2 in Figure 2D.

(4) Was CENPA enriched in the centromere DNA-O-MAP? If not, have the authors scaled up (e.g., with ten times more cells) to see if the local proteome becomes deeper and detects relevant low-abundance proteins like CENPA or HJURP? This would be very informative.

We did not observe CENPA, and we had originally contemplated the experiment the reviewer suggested, but noted that CENPA has only two tryptic peptides (>7 AA, <35AA), and they are both in the commonly phosphorylated region of the protein. Rather than scale up these experiments, we decided to attempt DNA OMAP on the non-repetitive locus experiments.

(5) Using a few million cells, I do not see how the starting chromatin amount could range from 0.5 to 7 mg, as shown in Figures 2 and 3. How were these figures calculated? One diploid cell contains approximately 6 pg of DNA/chromatin, which means one billion cells represent about 6 mg of DNA/chromatin (a typical measurement for these methods).

Thanks to the reviewer for catching this, that should have been the total lysate amount, not chromatin mass. We have corrected Figures 2 and 3.

(6) Figure S1: There is no indication of the metrics used for the shades of red.

We have added a gradient legend to depict this.

(7) What is the purpose of HCl in the experiment?

HCl treatment was done to reduce autofluorescence for imaging (PMID: 39548245).

(8) I could not find the MS dataset on the server using the provided accession number (PDX054080).

Thank you for pointing this out, we have confirmed the dataset is public now and added the new datasets for the Xi/Xa and Hox studies. We also note that the accession should be “PXD054080”

(9) Why desthiobiotin instead of biotin?

We have tested both; desthiobiotin was helpful to reduce adsorption to surfaces. Either biotin or desthiobiotin can be used, though, for OMAP.